**EMBO** *reports*

# IMPDH2 filaments protect from neurodegeneration in AMPD2 deficiency

Marco Flores-Mendez [1,2], Laura Ohl [1,2], Thomas Roule [1,2], Yijing Zhou[1,2], Jesus A Tintos-Hernández[3], Kelsey Walsh[1,2], Xilma R Ortiz-González [3,4] & Naiara Akizu [1,2]✉

## Abstract

Metabolic dysregulation is one of the most common causes of pediatric neurodegenerative disorders. However, how the disruption of ubiquitous and essential metabolic pathways predominantly affect neural tissue remains unclear. Here we use mouse models of a childhood neurodegenerative disorder caused by AMPD2 deficiency to study cellular and molecular mechanisms that lead to selective neuronal vulnerability to purine metabolism imbalance. We show that mouse models of AMPD2 deficiency exhibit predominant degeneration of the hippocampal dentate gyrus, despite a general reduction of brain GTP levels. Neurodegeneration-resistant regions accumulate micron-sized filaments of IMPDH2, the rate limiting enzyme in GTP synthesis, while these filaments are barely detectable in the hippocampal dentate gyrus. Furthermore, we show that IMPDH2 filament disassembly reduces GTP levels and impairs growth of neural progenitor cells derived from individuals with human AMPD2 deficiency. Together, our findings suggest that IMPDH2 polymerization prevents detrimental GTP deprivation, opening the possibility of exploring the induction of IMPDH2 assembly as a therapy for neurodegeneration.

**Keywords** Impdh2 Filaments; Ampd2; Metabolons; Purine Nucleotides; Pontocerebellar Hypoplasia
**Subject Categories** Metabolism; Molecular Biology of Disease; Neuroscience

## Introduction

Protein aggregation is a hallmark of age-related neurodegenerative disorders that leads to a progressive loss of vulnerable neurons (Taylor et al, 2002). However, neurodegeneration can also occur in the absence of protein aggregation. Metabolic dysregulation, for example, is a common cause of neurodegeneration, especially in pediatric populations (Pierre, 2013). A significant proportion of inherited metabolic disorders show progressive decline of neurologic, cognitive, and motor functions as central signs of neurodegeneration (Saudubray and Garcia-Cazorla, 2018; Wong, 1997). In these conditions, neurodegeneration is associated with the accumulation of toxic metabolites, depletion of key metabolites, or bioenergetic defects.

Disruption of purine nucleotide metabolism underlies the pathogenesis of a large group of inherited metabolic disorders caused by mutations in key enzymes of the purine nucleotide synthesis pathway (Balasubramaniam et al, 2014). Purine nucleotides (i.e., ATP and GTP) supply cells with energy, constitute building blocks of DNA and RNA, and participate in intra- and inter-cellular signaling, therefore supporting cellular proliferation, survival, and metabolic needs. To meet increased purine demands, proliferative cells synthetize purine nucleotides de novo in serial steps that convert phosphoribosyl pyrophosphate (PRPP) to inosine monophosphate (IMP) (Hartman and Buchanan, 1959; Lane and Fan, 2015; Watts, 1983). IMP is then used as common intermediate for de novo synthesis and interconversion of adenine and guanine nucleotides (i.e., ATP, ADP, AMP and GTP, GDP and GMP) (Fig. 1A). In contrast, postmitotic cells favor the salvage pathway, which recycles free purine bases into purine nucleotides (Hartman and Buchanan, 1959; Lane and Fan, 2015; Watts, 1983). The two pathways involve a series of sequential and often bidirectional reactions precisely coordinated by key metabolic enzymes.

The assembly of purine metabolism enzymes into micron-sized structures is emerging as a mechanism to fine tune and compartmentalize purine nucleotide biosynthesis within cells. For example, upon purine nucleotide starvation, up to 9 enzymes of the de novo purine biosynthetic pathway coordinate their activities by assembling into dynamic, microscopically visible intracellular metabolons called purinosomes (An et al, 2008). These purinosomes interact with mitochondria to promote de novo purine biosynthesis in response to increased local demand for purine nucleotides (French et al, 2016; Pareek et al, 2020). In addition, IMPDH1 and IMPDH2, the rate limiting enzymes in the de novo biosynthesis of guanine nucleotides, form reversible and microscopically visible filament bundles, also called rods and rings or cytoophidia (Carcamo et al, 2011; Gunter et al, 2008; Ji et al, 2006;

[1]Perelman Center for Cellular and Molecular Therapeutics, Children's Hospital of Philadelphia, Philadelphia, PA, USA. [2]Department of Pathology and Laboratory Medicine, Perelman School of Medicine, University of Pennsylvania, Philadelphia, PA, USA. [3]Division of Neurology and Center for Mitochondrial and Epigenomic Medicine, The Children's Hospital of Philadelphia, Philadelphia, PA 19104, USA. [4]Department of Neurology, Perelman School of Medicine, University of Pennsylvania, Philadelphia, PA 19104, USA. ✉E-mail: aquizun@chop.edu

    

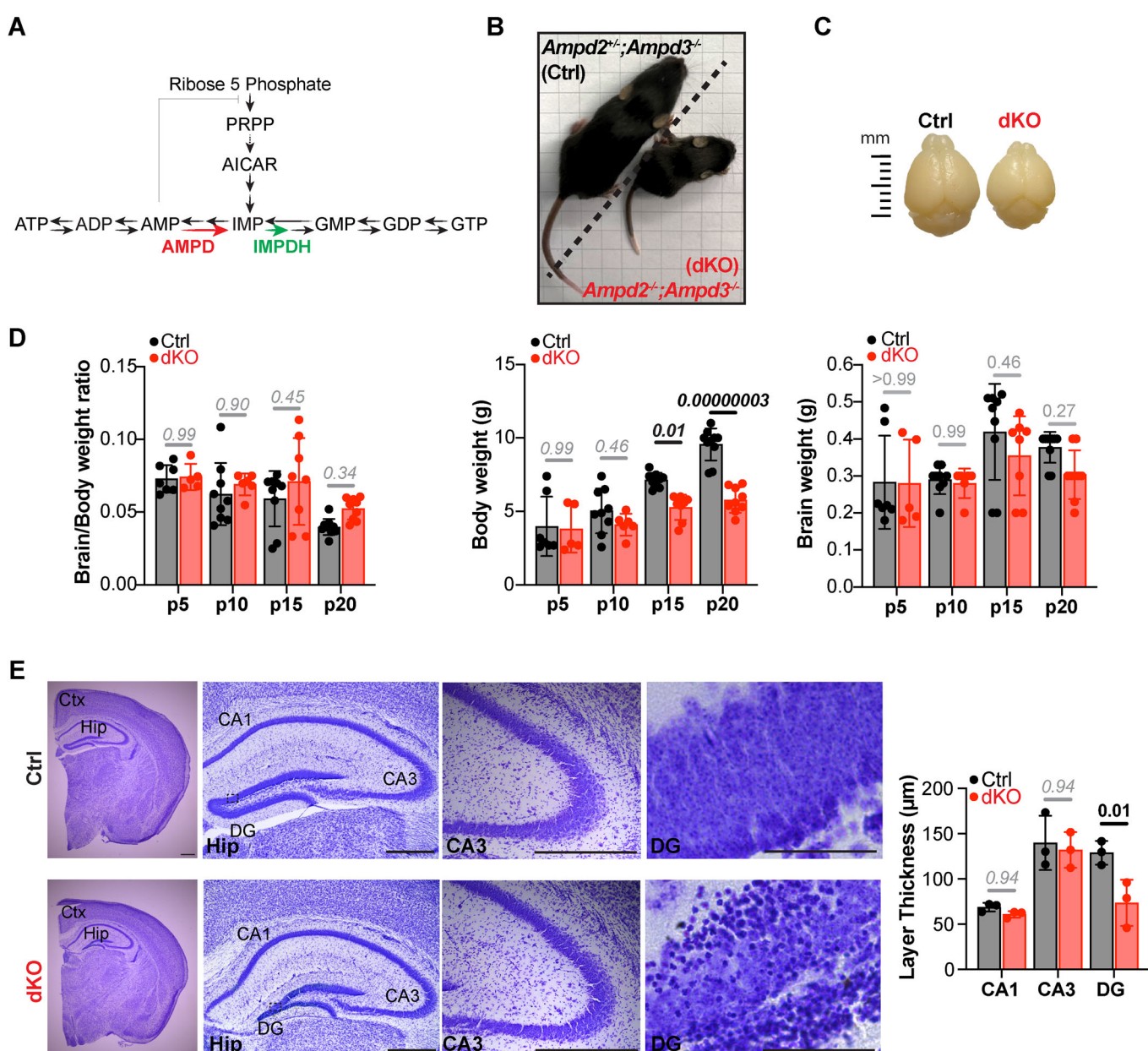

**Figure 1. Ampd2 and Ampd3 double knockout (dKO) mice show selective hippocampal neurodegeneration.**

(A) Schematic representation of the nucleotide metabolic pathway. The red and green arrows indicate enzymatic reaction catalyzed by AMPD2 and IMPDH2 respectively. (B) Representative image of control (Ctrl) and double knockout (dKO) mice at postnatal day 20 (p20). (C) Brain images of Ctrl and dKO mice at p20. (D) Bar graphs showing mean ± SD of brain/body weight ratio (left), body weight (middle), and brain weight (right) of $n = 5$–10 control and dKO mice at postnatal days 5, 10, 15, and 20. Each dot in the graphs represents data from one mouse (Ctrl: p5 $n = 7$, p10 and p15 $n = 9$, p20 $n = 10$; dKO: p5 $n = 5$, p10 $n = 6$, p15 $n = 8$, p20 and $n = 9$). Significance was calculated using two-way ANOVA with Sidak's post hoc analysis for multiple comparison. (E) Representative Nissl staining of forebrain coronal sections from Ctrl and dKO mice at p20 showing reduced thickness and pyknotic cells in hippocampal DG region of dKO mice. Graph shows mean ± SD of CA1, CA3, and DG layer thickness in $n = 3$ mice per genotype. Significance was calculated using two-way ANOVA with Sidak's post hoc analysis for multiple comparison. Ctx = Cortex; Hip = Hippocampus; CA3 = Cornu Ammonis; DG=Dentate Gyrus. Scale bars: whole brain, Hip, and CA3, 500 μm; Scale bars: DG, 50 μm. Source data are available online for this figure.

Juda et al, 2014). The effect of these micron-scale filament bundles on IMPDH1/2 enzymatic function is likely complex, but recent evidence show that, in vitro, individual filaments protect IMPDH1/2 from allosteric inhibition by GTP (Fernandez-Justel et al, 2019; Johnson and Kollman, 2020). Consistently, in vivo, IMPDH1/2 assembles at conditions that require expansion of guanine nucleotide pools, like in activating immune cells (Calise et al, 2018; Duong-Ly et al, 2018) or highly proliferating cells (Ahangari et al, 2021; Carcamo et al, 2011; Keppeke et al, 2018). Furthermore, micron-sized IMPDH1/2 filaments are formed under nutrient depriving conditions (Calise et al, 2014; Calise et al, 2016) or pharmacologic inhibition of nucleotide enzymes (Carcamo et al,

2011; Ji et al, 2006). Nevertheless, the implication of these macromolecular assemblies in physiological and pathological conditions remains poorly understood.

The nervous system is particularly vulnerable to imbalances of purine nucleotide and nucleoside pools given their additional functions as neuromodulators, neurotransmitters, and secondary messengers (Badimon et al, 2020; Pascual et al, 2005). Accordingly, several inherited purine nucleotide metabolism disorders show neurologic manifestations (Camici et al, 2010). Motor and cognitive disability and self-injurious behavior are hallmarks of Lesch-Nyhan syndrome (LNS) caused by mutations in *Hypoxanthine Guanine Phosphoribosyltransferase* (*HPRT1*), a key enzyme in the purine nucleotide salvage pathway. HPRT1 deficiency impairs the recycling of purine bases into purine nucleotides and leads to an overproduction of uric acid. However, treatments to reduce uric acid accumulation do not improve neurological phenotypes, which suggest an alternative neuropathogenic mechanism (Fu et al, 2015; Lesch and Nyhan, 1964). Likewise, overproduction of uric acid is a hallmark of PRPS1 superactivity disorder, characterized by gout which occasionally co-occurs with sensorineural deafness, cognitive deficits, and hypotonia (Becker et al, 1988; Sperling et al, 1972). In contrast, loss of function mutations in *PRPS1*, which cause a spectrum of diseases with clinical features of diverse severity, including neurodegeneration, likely results from depletion of purine biosynthesis as indicated by reduced levels of purines and uric acid in urine and plasma of patients (Arts et al, 1993; de Brouwer et al, 2007; Kim et al, 2007; Synofzik et al, 2014). Furthermore, heterozygous mutations in *IMPDH2* have recently been associated with neurodevelopmental disorders and mutations in *IMPDH1* lead to common retinal degenerative disorders (i.e., retinitis pigmentosa and leber congenital amaurosis) (Bowne et al, 2002; Kennan et al, 2002). Mutations in *IMPDH1/2* alter the conformation of their filamentous assemblies, potentially leading to a dysregulation of the guanine biosynthetic pathway as a disease mechanism (Buey et al, 2015; O'Neill et al, 2023). While metabolic consequences of mutations in purine nucleotide enzymes have been extensively investigated, they usually do not explain the pathogenesis of neurologic manifestations. Furthermore, disease specific variability in clinical manifestations suggests tissue specific regulatory mechanism and vulnerabilities that remain unclear.

Inactivating mutations in *AMPD2* cause Pontocerebellar Hypoplasia type 9 (PCH9) (Akizu et al, 2013; Kortum et al, 2018). Pontocerebellar hypoplasias are a group of rare monogenic disorders characterized by a reduced volume of the brainstem and cerebellum, with variable involvement of other brain structures, often caused by fetal onset neurodegeneration. Patients usually present developmental delay, feeding problems and motor abnormalities, with the most severe cases showing regression and early death due to respiratory problems (van Dijk et al, 2018). Current classification comprises 17 types of PCH which are distinguished by the genetic diagnosis, brain imaging, and clinical features (Zakaria et al, 2024). Functional annotation of PCH associated genes suggest impaired protein synthesis, RNA or energy metabolism as the underlaying pathogenic mechanism (van Dijk et al, 2018). PCH type 9 (PCH9) is clinically distinguishable from the other PCH types by the 'figure 8' shape of the midbrain in axial brain images (Akizu et al, 2013). Additional clinical features of PCH9 include, global developmental delay, postnatal microcephaly, atrophy of the cerebral cortex and thinner corpus callosum (Akizu

et al, 2013; Kortum et al, 2018). As exception, patients carrying a homozygous null mutation that affects only one of the *AMPD2* isoforms show intact brain structure but early onset upper motoneuron degenerative disorder classified as hereditary spastic paraplegia 63 (HSP63) (Novarino et al, 2014).

*AMPD2* is one of three mammalian adenosine monophosphate (AMP) deaminase paralogs involved in the conversion of AMP to inosine monophosphate (IMP), a key enzymatic step for guanine nucleotide synthesis (i.e., GTP) within the purine metabolism pathway. While *AMPD2* is ubiquitously expressed, the other two paralogues, *AMPD1* and *AMPD3* are restricted to specific cell types or lowly expressed. *AMPD1* is predominantly expressed in the muscle, and homozygous mutations result in exercise-stress-induced muscle weakness and cramping (Fishbein et al, 1978), whereas *AMPD3* is predominantly expressed in erythrocytes, and homozygous mutations result in asymptomatic erythrocytic AMP accumulation (Zydowo et al, 1989). We previously showed that mutations in AMPD2 block the de novo and salvage production of GTP in PCH9 patient-derived neural progenitor cell (NPC) cultures (Akizu et al, 2013). Partial loss of brain GTP levels and neurodegeneration were recapitulated in AMPD deficient mice only upon the double deletion of *Ampd2* and *Ampd3*. Furthermore, AMPD deficiency in mice led to premature death before postnatal day 21 preceded by signs of selective hippocampal neurodegeneration. This difference with PCH9 patients, who show a predominant involvement of the midbrain and cerebellum, suggests species-specific vulnerabilities to selective neuronal degeneration by AMPD deficiency. However, mouse hippocampal neurodegeneration is a useful model to study cellular and molecular features that mediate vulnerability to AMPD deficiency.

Here, we show that selective neuronal vulnerability to AMPD deficiency is inversely correlated with buildup of micron-sized IMPDH2 filaments. Taking advantage of AMPD deficient mice we uncover a selective accumulation of IMPDH2 filaments in the hippocampal CA1-3 while the dentate gyrus shows neurodegeneration, neuroinflammation and lack of IMPDH2 filaments. To assess the effect of IMPDH2 aggregation in neurodegeneration further, we generated a forebrain specific AMPD-deficient mouse strain that survives to adulthood. Longitudinal histological analyses revealed a severe neurodegeneration of the hippocampal dentate gyrus while CA1-3 regions were free of reactive microglia and did not degenerate. Remarkably, CA1-3 regions of the hippocampus persistently showed an accumulation of IMPDH2 filaments, while these were sparse and thin in the dentate gyrus. Finally, we demonstrate that PCH9 patient-derived NPCs also accumulate IMPDH2 filaments and their disassembly compromises PCH9 NPC growth. Altogether our work suggests induction of IMPDH2 filament assembly as a potential therapeutic intervention for PCH9 associated neurodegeneration.

# Results

## AMPD deficiency leads to selective hippocampal neurodegeneration in mice

Unlike *AMPD2* deficiency in humans, which causes PCH9, *Ampd2* deficient mice are behaviorally and neuropathologically indistinguishable from their control littermates likely due to compensation

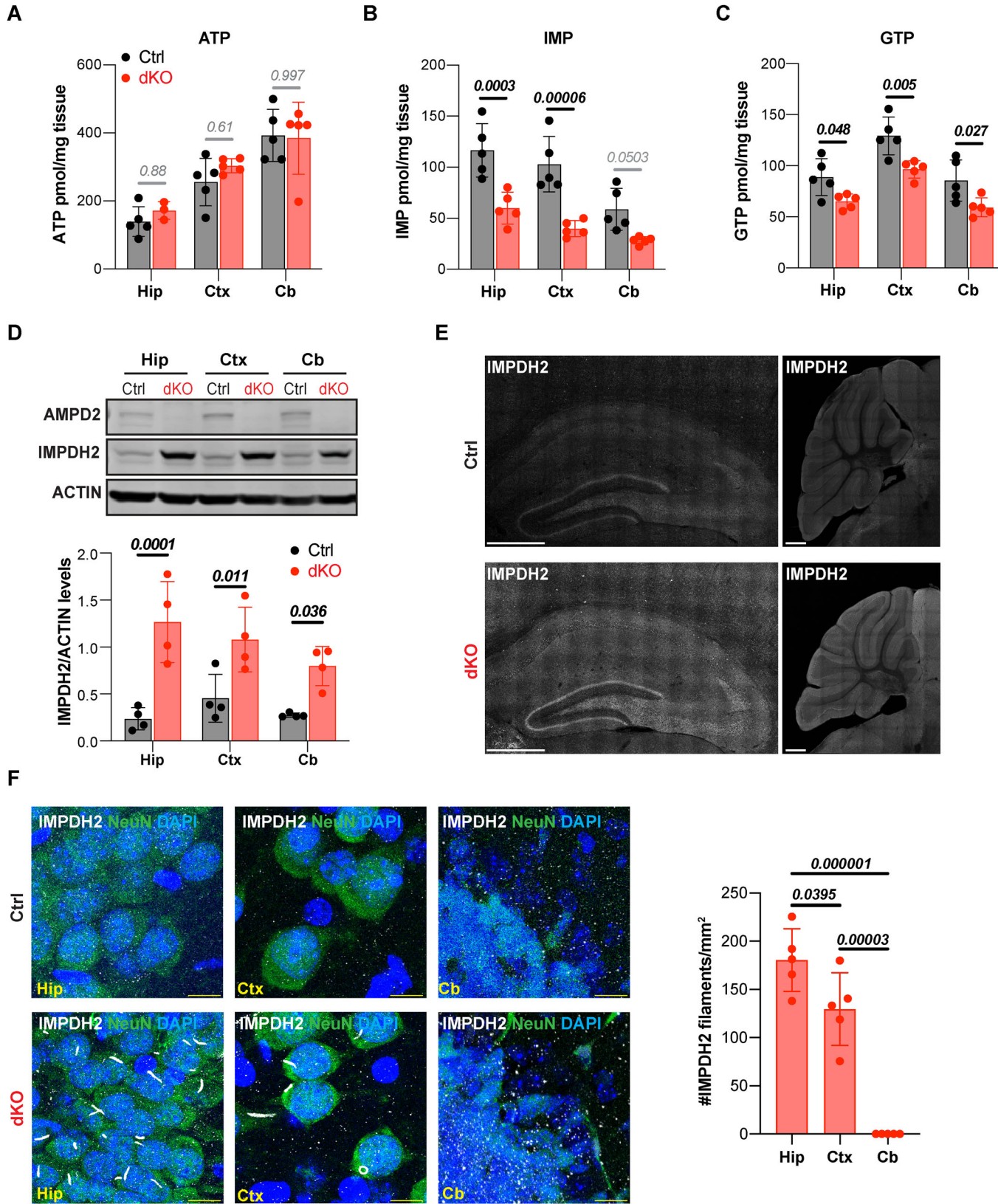

**Figure 2.  Hippocampal neurodegeneration is associated with IMPDH2 filament accumulation in dKO mice.**

(A–C) Bar graphs showing intact ATP levels and reduced IMP and GTP in Hippocampus (Hip), Cortex (Ctx), and Cerebellar (Cb) tissue of double knockout (dKO) mice compared to controls (Ctrl) at postnatal day 20 (p20). Bar graphs show mean ± SD of $n = 5$ animals per genotype. Significance was calculated using one-way ANOVA with Sidak's post hoc analysis for multiple comparison. (D) Representative Western Blot analysis showing absence of AMPD2 and upregulation of IMPDH2 in dKO compared to control mice at p20. ACTIN is shown as loading control. Graph shows mean ± SD of IMPDH2 band densitometry quantification relative to ACTIN in $n = 4$ animals per genotype. Significance was calculated using two-way ANOVA with Sidak's post hoc analysis for multiple comparison. (E) Representative immunostainings of IMPDH2 show an increase of IMPDH2 signal in Hip (left panel), Ctx (left panel), and Cb (right panel) of dKO mouse compared to controls at p20. Scale bars, 500 μm. (F) Representative immunostainings of IMPDH2 and NEUN (neuronal marker) show IMPDH2 filaments in neurons of dKO hippocampus and cortex but not in the cerebellum or in control mouse tissue. Scale bars, 10 μm. Graph shows mean ± SD of IMPDH2 filament density in the hippocampus (Hip), Cortex (Ctx), and Cerebellum (Cb) of p20 dKO mice. $n = 5$ mice per genotype. Significance was calculated using one-way ANOVA with Sidak's post hoc analysis for multiple comparison. Source data are available online for this figure.

by brain *Ampd3* expression (Akizu et al, 2013; Toyama et al, 2012). However, we previously showed that the homozygous deletion of both *Ampd2* and *Ampd3* (*Ampd2*$^{-/-}$;*Ampd3*$^{-/-}$, hereafter referred as dKO) leads to premature death at postnatal day 21 and signs of neurodegeneration, particularly affecting the hippocampus. To further determine the extent of neurodegeneration in dKO mice, we performed several morphological and histological analysis. As previously shown dKO mice were born at expected Mendelian ratio and survived up to postnatal day 21 (p21) showing weakness, motor difficulties and growth restrictions as indicated by the smaller body size than their control littermates (Fig. 1B). Although with some variability, dKO mice also showed smaller brains at postnatal day 20 (p20) (Fig. 1C). To assess if the smaller brain was part of the growth impairment or the result of a global neurodegeneration, we monitored the progression of brain/body weight over the first 20 days of life in dKO and control littermates. Results showed that the brain weight is proportional to the body weight (Fig. 1D), suggesting that smaller brain is the result of growth impairment rather than generalized neurodegeneration.

Nevertheless, since neurodegeneration is a hallmark of PCH9 and signs of neuronal death were previously observed in AMPD deficient mice (Akizu et al, 2013), we proceeded to analyze dKO mouse brains at histologic level. We chose to start the analysis with mice at the oldest age (p20) to enhance the likelihood of detecting neurodegeneration. Histologic analysis showed structurally and histologically normal cerebellum and cerebrum in dKO mice (Fig. EV1A,B). However, in agreement with our previous work, dKO mice showed hippocampal deficiencies, particularly a thinner dentate gyrus with presence of pyknotic cells (Fig. 1E), suggesting that AMPD deficiency in mice leads to selective hippocampal neurodegeneration.

## AMPD deficiency induces IMPDH2 aggregation in the hippocampus

Having established that AMPD deficiency in mice leads to a predominant neurodegeneration of the hippocampus, we next sought to determine the causes of hippocampal vulnerability. AMPD2 deficiency blocks AMP to IMP conversion, which leads to a depletion of GTP worsened by the inhibition of the de novo purine biosynthetic pathway by adenine nucleotide accumulation (Akizu et al, 2013). Thus, we hypothesized that selective hippocampal vulnerability in dKO mice may result from a more severe purine nucleotide imbalance than in other brain regions. To test this hypothesis, we isolated hippocampi, cerebral cortices, and

cerebella from dKO mice and control littermates at p20 and analyzed purine nucleotide levels by LC/MS (Dataset EV1 and Appendix Fig. S1). Results showed nearly intact ATP levels in all the dKO brain regions analyzed (Fig. 2A), but an accumulation of AMP in the dKO hippocampus and cerebellum, and ADP in the dKO cortex and cerebellum (Fig. EV2A). In contrast, IMP, which is the direct product of AMP deamination by AMPD enzymes, was among the most reduced nucleotide in all analyzed dKO brain areas (Fig. 2B). Likewise, total guanine nucleotides, which are produced from IMP by IMPDH2, were similarly reduced in dKO hippocampus, cortex and cerebellum, with the exception of GDP which was lower only in dKO hippocampus and preserved in cortex and cerebellum (Figs. 2C and EV2B).

To further validate nucleotide deficits in dKO mice with an orthogonal approach, we took advantage of the regulation of IMPDH2 expression by guanine nucleotides. In yeast and mammalian cell cultures, depletion of guanine nucleotide pool triggers the overexpression of IMPDH2 and their homologs (Escobar-Henriques and Daignan-Fornier, 2001; Zimmermann et al, 1998). In agreement, Western Blot (WB) analysis showed that reduced guanine nucleotides in dKO mice brain regions are associated with an increase of IMPDH2 protein levels (Fig. 2D). To assess if IMPDH2 upregulation was homogeneous or cell type specific, we performed IMPDH2 immunofluorescence analysis in brain sections and confirmed an homogeneous increase of IMPDH2 expression in all dKO brain regions (Fig. 2E). Remarkably, the immunofluorescence analysis uncovered micron-sized filamentous IMPDH2 structures in dKO mice brain sections but not in controls (Fig. 2E). However, despite the similarity in guanine nucleotide depletion and IMPDH2 overexpression between all analyzed brain regions, IMPDH2 filaments were most abundant in the hippocampus, particularly in CA1-3 regions, while only few were detected in the cortex and nearly none in the cerebellum (Fig. 2F). Accordingly, IMP to GTP ratio, which plays an important role on IMPDH2 filament assembly (Johnson and Kollman, 2020), was also the largest in the hippocampus (Fig. EV2C). Most of the IMPDH2 filaments had a rod shape, while fewer were arranged as a ring (Fig. 2F). Despite their similarity with primary cilia, co-immunostaining with acetylated tubulin confirmed that IMPDH2 filaments do not overlap with primary cilia (Fig. EV2D). Instead, by ultrastructure analysis, we detected a perinuclear enrichment of IMPDH2 filaments often showing mitochondria in the proximity (Fig. EV2E). Overall, these data show that the predominant vulnerability of the hippocampus to GTP depletion is associated with a disproportionate IMPDH2 filament assembly.

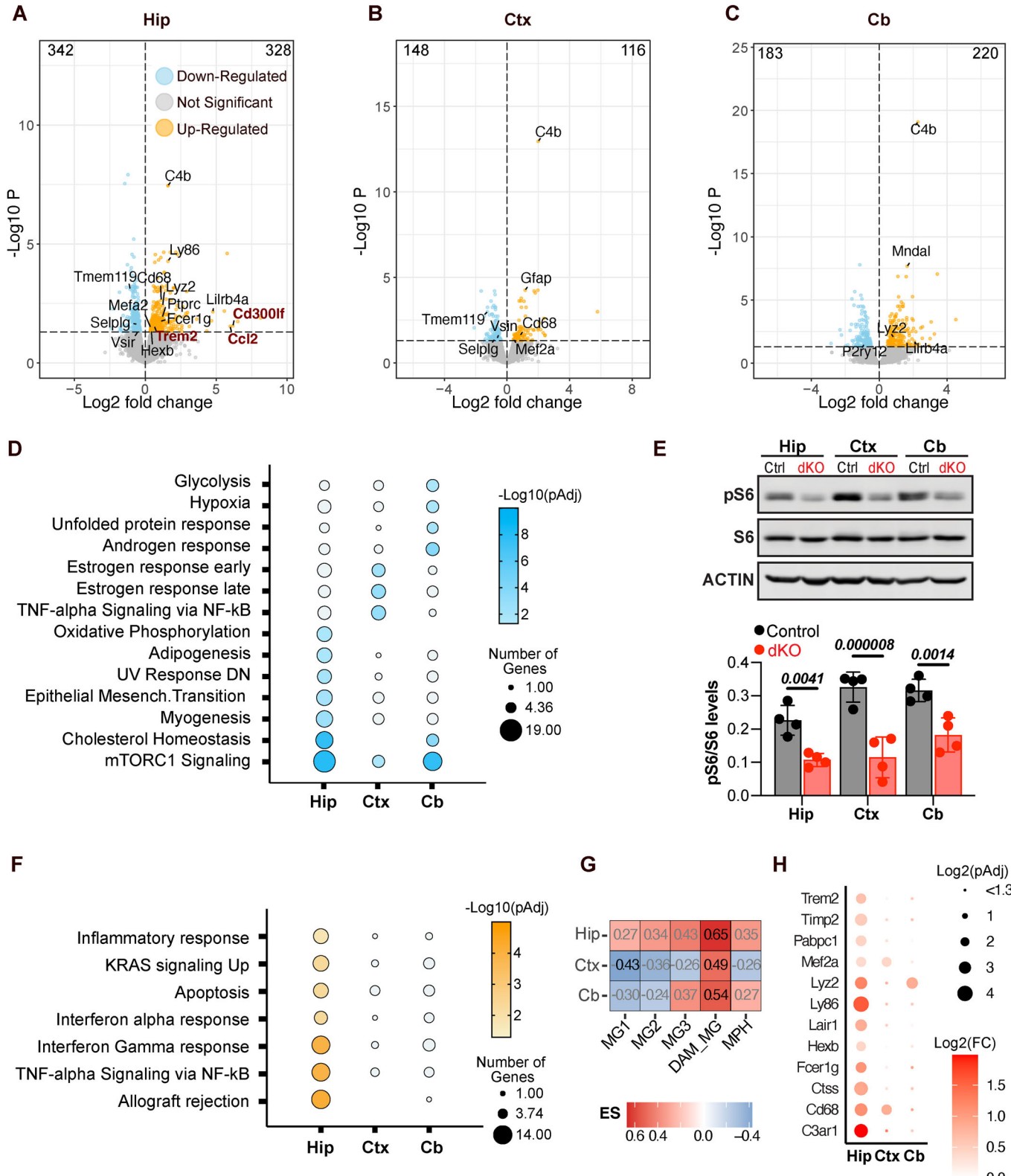

**Figure 3.** dKO brains show generalized mTORC1 signaling inhibition and hippocampal selective neurodegeneration and neuroinflammation transcriptomic signatures.

(A–C) Volcano plots of differentially expressed genes (DEGs) in Hippocampus (A), Cortex (B), and Cerebellum (C) of dKO vs Ctrl mice at postnatal day p20 ($n = 3$ mice per genotype). Horizontal dashed lines indicate statistical significance cut off ($padj < 0.05$ or $-\log10P > 1.301$). Number of significantly down- and up-regulated genes are indicated in the top left and right corners, respectively. Significance was calculated using the Wald Test followed by Benjamini–Hochberg method for correction of multiple comparisons. (D) Dot plot showing the functional annotation analysis of significantly downregulated genes in dKO vs Ctrl Hippocampus (Hip), Cortex (Ctx), and Cerebellum (Cb). White circles are non-significant ($-\log10(pAdj) < 1.301$). Significance was calculated using the Fisher's exact test followed by the Benjamini–Hochberg method for correction of multiple comparisons in Enrichr. (E) Western Blot of pS6 and S6 in dKO Hip, Ctx, and Cb. ACTIN is shown as loading control. Graph shows mean ± SD of pS6 band densitometry quantification relative to total S6 in $n = 4$ animals per genotype. Significance was calculated with two-way ANOVA and Sidak's post hoc analysis for multiple comparison. (F) Dot plot showing the functional annotation analysis of significantly upregulated genes in dKO vs Ctrl Hip, Ctx, and Cb. White circles are non-significant ($-\log10(pAdj) < 1.301$). Significance was calculated using the Fisher's exact test followed by the Benjamini–Hochberg method for correction of multiple comparisons in Enrichr. (G) Heatmap of GSEA enrichment scores (ES) for microglia gene signatures in dKO vs Ctrl Hip, Ctx, and Cb DEGs. MG1 = microglia type 1, MG2 = microglia type 2, MG3 = microglia type 3, DAM_MG = Disease associated microglia, MPH = Macrophage. Significance of ES was calculated using GSEA permutation adjusted with the Benjamini–Hochberg method for multiple comparison. ES of signatures with pAdj < 0.05 were marked in black inside each rectangle of the heatmap and those with pAdj > 0.05 in gray. (H) Dot plot showing the log-transformed fold change in expression and adjusted $p$-value of selected microglia gene markers between dKO and Ctrl in Hippocampus, Cortex, and Cerebellum. Smaller dots represent non-significant changes. Significance was calculated using the Wald Test followed by Benjamini–Hochberg method for correction of multiple comparisons. Source data are available online for this figure.

## Brain region specific transcriptomes reveal apoptosis and neuroinflammation signatures in dKO mice hippocampi

To determine why the hippocampus is more vulnerable to GTP depletion and IMPDH2 assembly in dKO mice, we conducted an mRNA sequencing (RNAseq) analysis of the three brain regions exhibiting different degrees of neurodegeneration and IMPDH2 aggregation: the hippocampus showing neurodegeneration and IMPDH2 filaments, the cerebral cortex with apparently no neurodegeneration and low density of IMPDH2 filaments, and the cerebellum with no degeneration or IMPDH2 filaments. We first analyzed the RNAseq data to identify differentially expressed genes (DEGs) between control and dKO mice in each brain region (Dataset EV2). Remarkably, the hippocampus exhibited the largest number of DEGs with 342 genes downregulated and 328 upregulated in dKO compared to control mice (Fig. 3A–C). Functional annotation analysis of the three brain regions also showed a significant enrichment of 'mTORC1 signaling' among downregulated DEGs in dKO mice, with 19 genes contributing to this category in the hippocampus, 6 in the cortex and 14 in the cerebellum (Fig. 3D). Interestingly, depletion of guanine nucleotides is known to inhibit mTORC1 activity to compensate for the high nucleotide demand of mTORC1-stimulated ribosomal RNA synthesis (Hoxhaj et al, 2017). Therefore, the downregulation of mTORC1 signaling genes may be a consequence of the overall guanine nucleotide reduction in dKO mice. To test for this possibility and asses if brain region specific mTORC1 inhibition could be associated with selective hippocampal neurodegeneration, we extracted proteins from the hippocampus, cortex, and cerebellum of control and dKO mice and analyzed the levels of ribosomal protein S6 phosphorylation by WB, as a proxy of mTORC1 signaling activity. Results confirmed mRNA-seq data and suggested that mTORC1 signaling is similarly inhibited in the three brain regions of dKO mice (Fig. 3E), ruling out mTORC1 inhibition as the underlying cause of the selective hippocampal neurodegeneration.

On the other hand, functional annotation analysis of upregulated DEGs revealed a significant enrichment of apoptosis associated genes only in the dKO hippocampus (Fig. 3F). This data is in line with the selective hippocampal neurodegeneration we found by histological analysis. Moreover, only DEGs upregulated in the hippocampus were enriched for inflammatory response,

interferon, and TNF alpha response pathway annotations, all associated with neuroinflammation (Fig. 3F). To further confirm that hippocampal selective neurodegeneration is associated with an inflammatory response, we conducted a gene set enrichment analysis (GSEA) of DEGs in the three brain regions using as input microglia specific gene signatures curated from published single-cell RNA sequencing datasets (Keren-Shaul et al, 2017). Interestingly, the dKO hippocampus showed the highest enrichment score for all microglia associated gene signatures, including significantly enriched disease associated microglia (DAM-MG) gene signature (Fig. 3G). Consistently, several microglia markers (i.e., *Trem2*, *Lyz2*, *Cd68*) were concurrently upregulated in the dKO hippocampus, but not in the dKO cortex and cerebellum (Fig. 3H). These data support the selective hippocampal neurodegeneration of dKO mice, and suggest that GTP depletion and IMPDH2 filament accumulation in the hippocampus may be associated with microglia accumulation.

## Microglia accumulation is restricted to regions of the hippocampus with low density of IMPDH2 filaments

IMPDH2 is well known for its role in supporting metabolic needs of immune system, particularly during T-cell stimulation (Duong-Ly et al, 2018; Zimmermann et al, 1998). Furthermore, recent data suggest that IMPDH2 may be involved in brain microglia infiltration (Liao et al, 2017). Based on these data, we considered the possibility that IMPDH2 filaments in dKO hippocampus may result from IMPDH2 assembly in infiltrating or reactive microglia. To test this hypothesis, we co-immunostained brain sections with anti-CD68 microglia marker and anti-IMPDH2. Results showed an accumulation of microglia only in the dKO the hippocampus (Fig. 4A), and predominantly in the DG (Fig. 4A right panel). This result is consistent with signs of neurodegeneration we previously observed in the DG but not in CA1-3 of the hippocampus (Fig. 1E), therefore supporting a selective vulnerability of the DG to degenerate under AMPD deficiency in mice. However, none of the CD68+ microglia had IMPDH2 filaments (Fig. 4B). Furthermore, we noticed that IMPDH2 filaments were most abundant in neurons of the CA1-3 regions, while barely detectable in the neurodegenerating DG of dKO mice (Figs. 4B and EV3A). These data suggest that the selective vulnerability of DG neurons to degeneration may be associated with their inability to assemble IMPDH2 filaments. Alternatively, the absence of IMPDH2 filaments in the degenerating DG could indicate

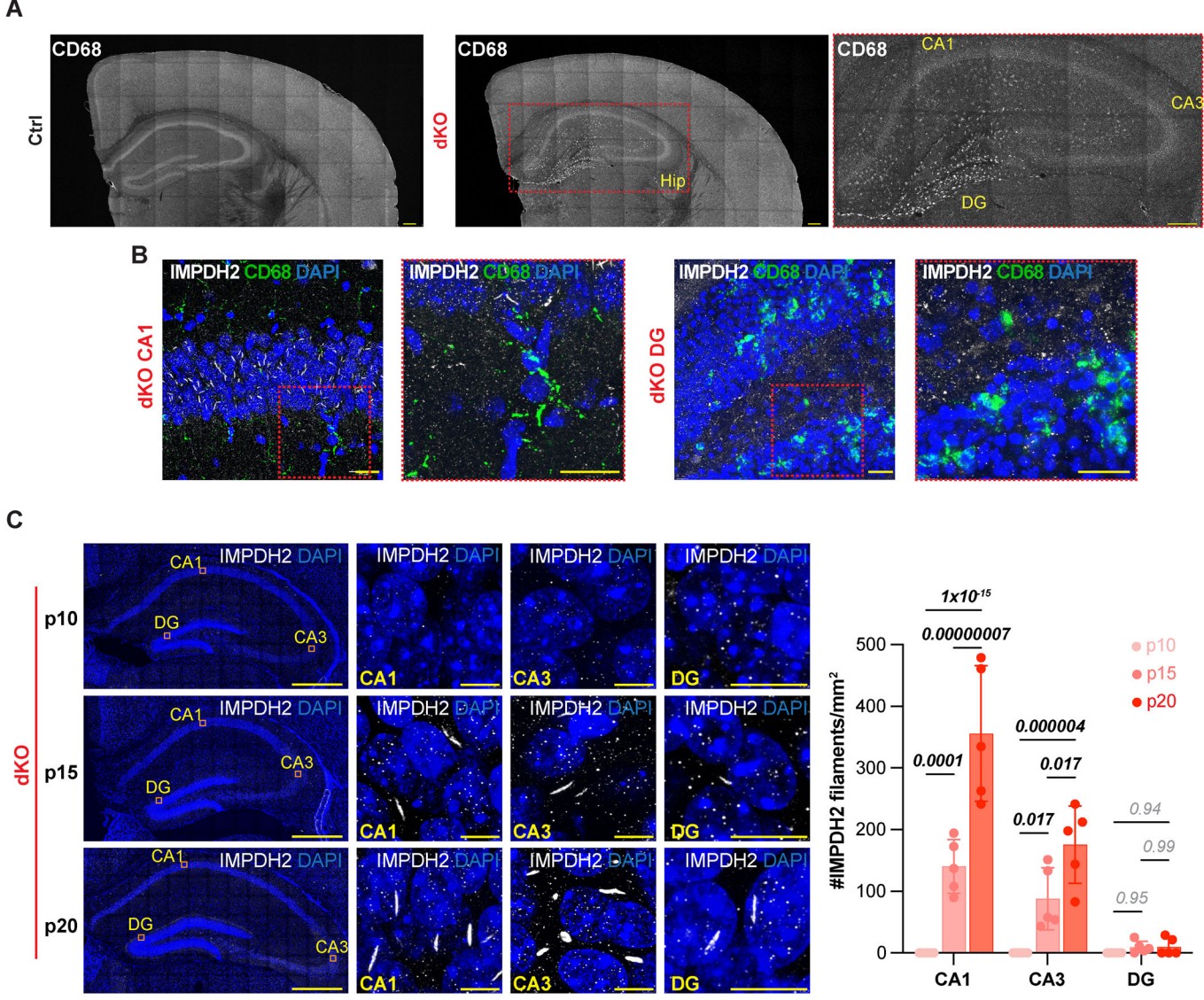

**Figure 4. IMPDH2 filaments and microglia accumulation are inversely correlated in dKO hippocampus.**

(A) Representative images of coronal brain sections from Control (Ctrl) and double knockout (dKO) mice at postnatal day 20 (p20) immunostained with anti-CD68 antibody. Inset magnification (red square) shows hippocampal DG regions with abundant CD68+ microglia in dKO mice. Scale bars, 200 μm. (B) Representative images of hippocampal CA1 and DG regions of dKO mice immunostained for IMPDH2 and CD68. Inset magnifications (red square) show CD68+ microglia in IMPDH2 filament dense hippocampal CA1 region (left), and IMPDH2 filament free DG region (right). Scale bars, 20 μm. (C) Representative anti-IMPDH2 immunostaining images of hippocampal regions (CA1, CA3, and DG) in dKO mice at postnatal days 10, 15, and 20 showing progressive accumulation of IMPDH2 filaments with higher density in CA1-3. Scale bars: whole Hip: 500 μm; Scale bars: CA1, CA3, and DG, 10 μm. Graph shows mean ± SD of IMPDH2 filament density in CA1, CA3, and DG of p10, p15, and p20 dKO mice. $n = 5$ mice per genotype. Significance was calculated using two-way ANOVA with Sidak's post hoc analysis for multiple comparison. CA=Cornu Ammonis; DG=Dentate Gyrus. Source data are available online for this figure.

that DG neurons containing IMPDH2 filaments had already been lost by the time we analyzed the brains at p20. To distinguish between these two possibilities, we conducted a time course analysis of IMPDH2 filament density in dKO brain sections, including younger mice. Results revealed complete absence of IMPDH2 filaments in dKO mice brains at p10 and a progressive accumulation from p15 to p20 in the cortex, and CA1 and CA3 regions of the hippocampus (Figs. 4C and EV3B). Although some IMPDH2 filaments were detected in the DG region of dKO mice at p15 and p20, their density was consistently low (Fig. 4C). Together, these data show that the lack

of IMPDH2 filaments is associated with the selective neurodegeneration of the DG and suggests that IMPDH2 filaments protect CA1-3 neurons from degeneration caused by AMPD deficiency.

## IMPDH2 filament accumulation in CA1-3 associates with resistance to neurodegeneration

To further test whether IMPDH2 filaments protect from neurodegeneration, we sought to assess if differences in survival and IMPDH2 density between neurons in CA1-3 and DG persisted in

older dKO mice. However, the premature death of dKO mice at p21 precluded us from assessing neurodegeneration further. To overcome this challenge, we sought to generate an AMPD-deficient mouse model that survived longer by deleting AMPD activity only in the forebrain. To this end, we generated a conditional *Ampd3* allele by removing the *LacZ* trapping cassette from the *Ampd3* knockout-first allele and subsequently breeding it into *Ampd2* knock out and *Emx1:Cre* knock in mice (Fig. EV4A). This strategy produced a forebrain specific AMPD-deficient mice, hereafter named cdKO. The upregulation of IMPDH2 in cerebral cortex and hippocampus observed by WB analysis confirmed the deletion of AMPD activity in forebrain regions (cortex and hippocampus) (Fig. EV4B). In contrast, IMPDH2 levels in the cdKO cerebellum were comparable to control mice (Fig. EV4B) as predicted by the lack of Emx1:cre expression in the cerebellum (and therefore intact cerebellar *Ampd3*).

Having confirmed the forebrain specific deletion of AMPD, we next characterized its effects on survival, body weight, and brain weight. As expected, due to the preserved AMPD activity in most body regions, the lifespan and body weight of cdKO mice were similar to those of their control littermates (Fig. EV4C,D). We only detected a mild reduction of brain weight at 8-week-old mice (Fig. EV4E), although brain to body weight ratio was similar between cdKO and control littermates (Fig. EV4F). To test if smaller brain was associated with progressive neurodegeneration, we analyzed brain sections histologically. Like dKO mice, 5-week-old cdKO mice exhibited thinner DG and preserved CA1-3 of the hippocampus (Fig. 5A). Remarkably, only the DG region showed a progressive tissue loss at 8 weeks of age, while CA1-3 regions remained similar to controls over time (Fig. 5B).

We next assessed whether DG degeneration in cdKO mice was associated with a selective microglia accumulation. Consistent with our findings in dKO mice, CD68+ microglia accumulation was predominant in hippocampal DG region of cdKO mice, while CA1-3 regions were only sparsely labeled with CD68+, including at older ages (Figs. 5C and EV4G). Furthermore, IMPDH2 immunofluorescence stainings showed low density and small IMPDH2 filaments in cdKO DG at 5 and 8 weeks of age (Fig. 5D,E). In contrast, microglia- and neurodegeneration-free CA1-3 regions persistently accumulated large and dense IMPDH2 filaments, at least up to 24 weeks of age (Figs. 5D,E and EV4H). Likewise, IMPDH2 filaments progressively accumulated in cdKO cortex (Fig. EV4H,I). Together, this data shows that IMPDH2 filament accumulation is associated with non-degenerating brain regions of AMPD deficient mice, suggesting that IMPDH2 filaments may protect from neurodegeneration.

## Protection by IMPDH2 filaments is conserved in human AMPD2 deficient patient derived neural cells

Our previous work uncovered that AMPD2-deficient PCH9 patients' neural progenitor cells (NPCs) are vulnerable to adenosine supplementation in culture (Akizu et al, 2013). However, under standard culture conditions, PCH9 NPCs grow similarly to controls, suggesting compensatory mechanisms that support cell growth and survival despite AMPD2 deficiency (Akizu et al, 2013). To test whether IMPDH2 filament assembly could be involved in the compensatory mechanism, we derived NPCs from PCH9 patient-induced pluripotent stem cells (iPSCs) and a healthy control family member. PCH9 NPCs showed expression of PAX6 and NESTIN neural progenitor markers and cell growth comparable to control NPCs (Fig. EV5A). Furthermore, as we predicted, anti-IMPDH2 immunostaining revealed IMPDH2 filaments in a small percentage of PCH9 NPCs (8.2% ± 2.3 SD), but not in controls (Fig. 6A). This result suggests that IMPDH2 filament assembly could be involved in supporting the survival and growth of PCH9 NPCs under standard culture conditions.

To test if IMPDH2 filaments have a protective effect on PCH9 NPCs, we took advantage of the non-polymerizing but catalytically active IMPDH2-p.Y12A variant previously identified (Anthony et al, 2017). Overexpression of this variant has a dominant negative effect on IMPDH2 polymerization, preventing endogenous IMPDH2 filament assembly, while maintaining catalytic activity (Anthony et al, 2017). Thus, we transduced PCH9 NPCs with lentiviruses carrying either wild type (WT) or p.Y12A IMPDH2 transgenes, both tagged with a MYC peptide. To control for baseline growth rate, we also included PCH9 NPCs transduced with FLAG peptide coding sequence. By WB analysis we confirmed that IMPDH2-WT and IMPDH2-p.Y12A were overexpressed at similar levels (Fig. 6B). Furthermore, as previously reported in other cell types (Anthony et al, 2017; Keppeke et al, 2018), IMPDH2-WT overexpression led to IMPDH2 filament polymerization in NPCs, while no filaments were detected in IMPDH2-p.Y12A NPCs (Fig. 6C). We then monitored the effect of IMPDH2-p.Y12A overexpression on PCH9 NPCs growth over 120 h. While we did not detect differences between FLAG and IMPDH2-WT overexpressing PCH9 NPCs, IMPDH2-p.Y12A overexpression significantly reduced the growth of PCH9 NPCs (Fig. 6D). These data indicate that exogenously inducing IMPDH2 polymerization does not provide a growth advantage, likely due to limiting availability of GTP precursors (i.e., IMP), while inhibiting IMPDH2 filament assembly by IMPDH2-p.Y12A overexpression is detrimental for PCH9 NPCs growth.

Given that IMPDH2 filament assembly is proposed as a mechanism to boost guanine nucleotide synthesis under demanding conditions (Johnson and Kollman, 2020), we anticipated that the growth disadvantage caused by IMPDH2 filament disassembly could be correlated with reduced guanine nucleotide levels in IMPDH2-p.Y12A overexpressing PCH9 NPCs. To test this hypothesis we analyzed purine nucleotide levels of FLAG, IMPDH2-WT, and IMPDH2-p.Y12A PCH9 NPCs (Dataset EV3 and Appendix Fig. S2). Although we did not detect statistically significant differences in adenine or guanine nucleotide levels between the three conditions, IMPDH2-p.Y12A PCH9 NPCs had the lowest total guanine nucleotide levels in average (Figs. 6E and EV5B). While technical variability is a potential driver of this result, it is also possible that global guanine nucleotide levels are not involved in the protection provided by IMPDH2 filaments. Alternatively, IMPDH2 disassembly may only affect subcellular nucleotide compartments or a small percentage of NPCs that contain IMPDH2 filaments, therefore rendering bulk guanine nucleotide extract of PCH9 NPC nearly intact. Together, our work indicates that IMPDH2 filament polymerization plays an important role protecting neurons from AMPD deficiency-associated degeneration, although the involvement of guanine nucleotide biosynthesis in this process remains to be determined.

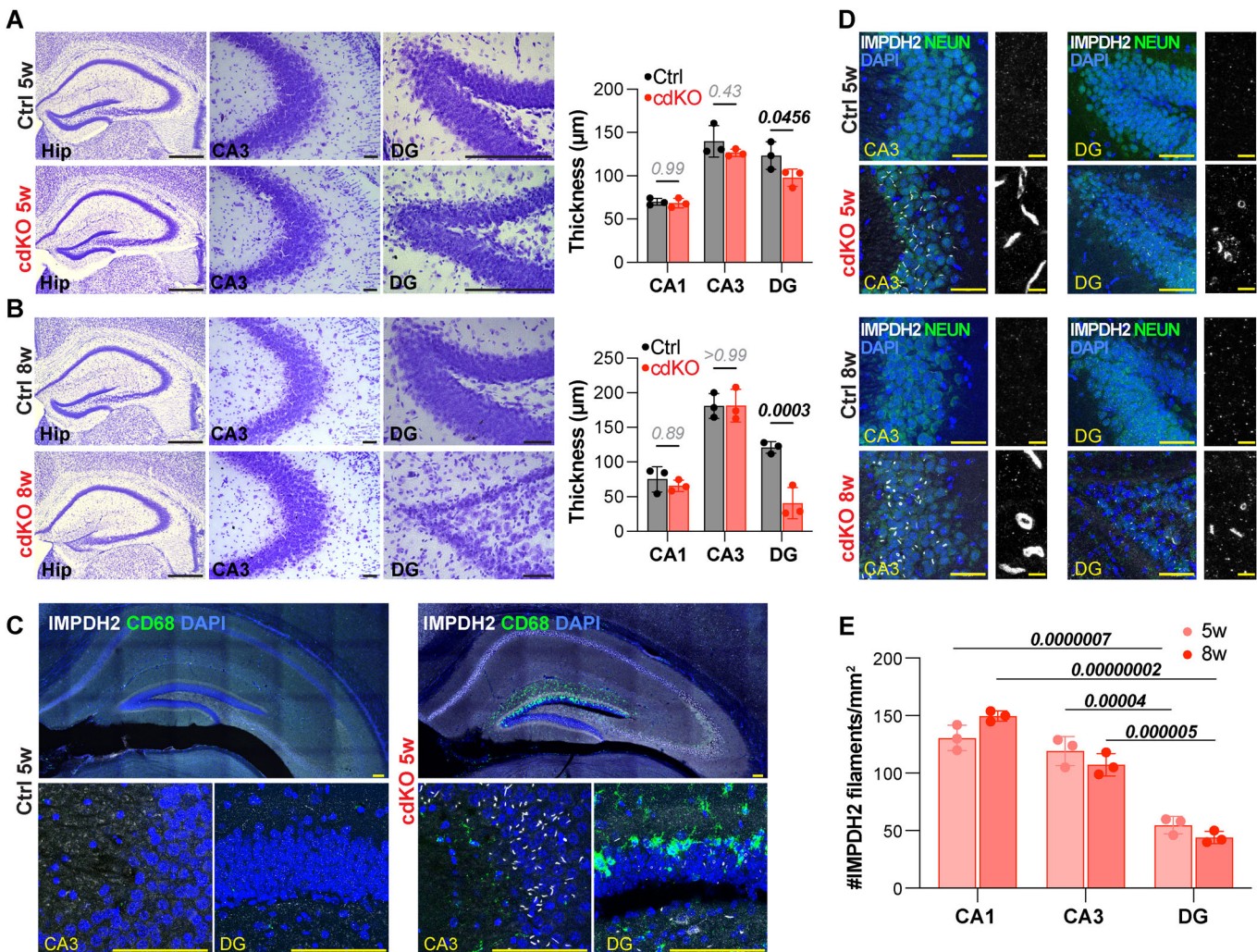

**Figure 5. IMPDH2 filaments associate with neurodegeneration resistant hippocampal regions in forebrain specific *Ampd2* and *Ampd3* knockout mice (cdKO).**

(A) Representative Nissl staining of brain sections showing full hippocampus and CA3 and DG regions in 5-week-old (5w) control (Ctrl) and conditional double knockout mice (cdKO). Pyknotic cells are present in cdKO DG region. Scale bars: whole Hip, 500 μm; Scale bars: CA3 and DG, 50 μm. Graph shows mean ± SD of CA1, CA3, and DG layer thickness in $n = 3$ mice per genotype. Significance was calculated using two-way ANOVA with Sidak's post hoc analysis for multiple comparison. (B) Representative Nissl staining of brain sections showing full hippocampus and CA3 and DG regions in 8-week-old (8w) Ctrl and cdKO mice. Images highlight thinner DG region in cdKO compared to Ctrl hippocampus. Scale bars: whole Hip: 500 μm; Scale bars: CA3 and DG, 50 μm. Graph shows mean ± SD of CA1, CA3, and DG layer thickness in $n = 3$ mice per genotype. Significance was calculated using two-way ANOVA with Sidak's post hoc analysis for multiple comparison. (C) Representative immunostainings of IMPDH2 and CD68 in 5-week-old (5w) control (Ctrl) and cdKO mouse hippocampus. Bottom panels show magnification of CA3 and DG region highlighting absence of CD68+ microglia and high density of IMPDH2 filaments in cdKO CA3, and high density of CD68+ microglia and low IMPDH2 filaments in cdKO DG region. Scale bars, 100 μm. (D) Representative immunostainings of IMPDH2 and NEUN (neuronal marker) in brain sections show persistent presence of IMPDH2 filaments in cdKO CA3 region and low density of small IMPDH2 filaments in cdKO DG. Scale bars, 50 μm (color panel) and 5 μm (gray scale panel). (E) Graph shows mean ± SD of IMPDH2 filaments density observed in hippocampal CA1, CA3, and DG regions of $n = 3$ cdKO mice at 5w and 8w of age. Significance was calculated using two-way ANOVA with Sidak's post hoc analysis for multiple comparison. Hip = Hippocampus; CA3 = Cornu Ammonis; DG = Dentate Gyrus. Source data are available online for this figure.

# Discussion

Selective neuronal vulnerability to metabolic dysfunction is a common feature of pediatric neurodegenerative disorders, but mechanisms that confer vulnerability or neuroprotection are largely unknown. Here we show that IMPDH2 polymerization into micron-sized filaments is associated with resistance to neuronal cell death in models of PCH9, a pediatric neurodegenerative disorder caused by AMPD2 deficiency. Remarkably, upregulation of IMPDH2, which often triggers its polymerization, is reported in

Alzheimer's diseases postmortem tissue and mouse models (Neuner et al, 2017; Puthiyedth et al, 2016; Xu et al, 2019), suggesting a broader implication of IMPDH2 polymerization in neurodegeneration. Therefore, our work rises the possibility for a role of IMPDH2 polymerization in neuroprotection and treatment of neurodegenerative disorders.

IMPDH2 catalyzes the rate limiting step for the biosynthesis of guanine nucleotides from IMP. Its activity is tightly regulated at multiple levels, including transcriptionally, posttranscriptionally, and allosterically by binding to purine nucleotides (Hedstrom,

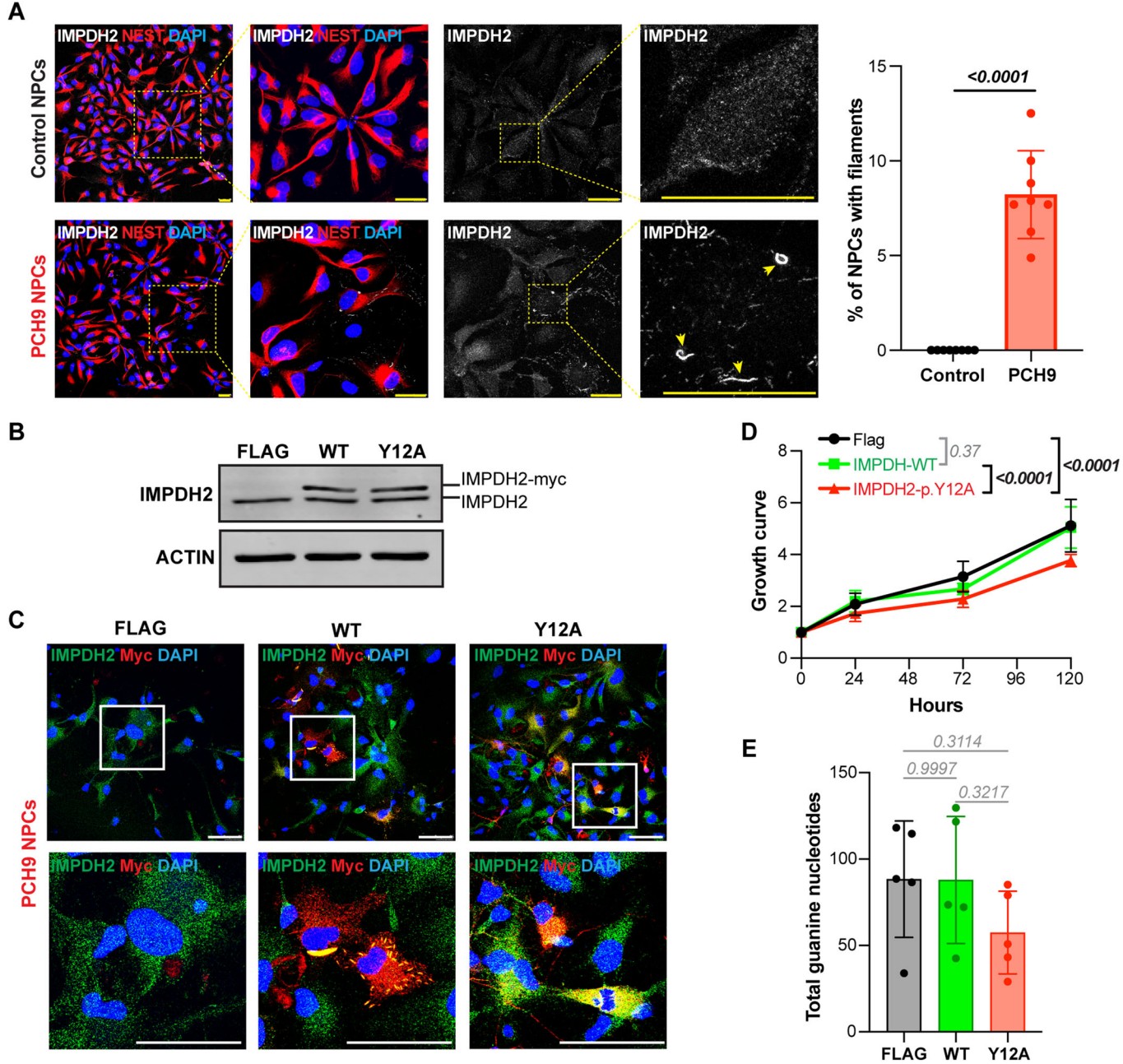

**Figure 6. IMPDH2 filaments protect PCH9 patient's NPCs from growth restrictions.**

(A) Representative immunostainings of IMPDH2 and NESTIN showing IMPDH2 filaments in PCH9 NPCs but not in controls. Yellow insets show IMPDH2 filament structures in PCH9 NPCs. Scale bars, 25 µm. Graph shows mean ± SD of $n = 8$ samples per group. (B) Western Blot analysis of IMPDH2 levels in PCH9 NPCs expressing FLAG (as control), IMPDH2-WT-myc (WT), or the non-polymerizing but catalytically active IMPDH2-p.Y12A-myc (Y12A) variant. (C) Representative immunostaining of IMPDH2 and MYC in PCH9 NPCs expressing FLAG, IMPDH2-WT-myc (WT), or IMPDH2-p.Y12A-myc (Y12A). Scale bars, 50 µm. (D) Growth curves quantified with MTT assay show slower growth in IMPDH2p.Y12A-myc expressing NPCs compared to FLAG and IMPDH2-WT-myc NPCs. Data represents mean ± SD of 6 independent cultures. Statistical difference was calculated comparing slops of simple linear regression with one-way ANOVA and Tukey's post hoc analysis for multiple comparison. (E) Guanine nucleotides levels in PCH9 NPCs expressing FLAG, IMPDH2-WT-myc (WT), or IMPDH2-p.Y12A-myc (Y12A). Graph shows mean ± SD of $n = 5$ samples per group. Significance was calculated with one-way ANOVA with Tukey's post hoc analysis for multiple comparison. Source data are available online for this figure.

2009). Cellular conditions that deprive guanine nucleotide levels induce IMPDH2 upregulation and conformational changes that boost enzymatic activity, which are reversed once GTP levels are restored (Hedstrom, 2009). Furthermore, IMPDH2 polymerization into micron-sized filaments is emerging as a mechanism that reduces the sensitivity of IMPDH2 to feedback inhibition by guanine nucleotides (Fernandez-Justel et al, 2019; Johnson and Kollman, 2020). Therefore, under conditions of high GTP demand, IMPDH2 assembly into filaments would enhance GTP synthesis above the homeostatic levels (Johnson and Kollman, 2020).

Accordingly, IMPDH2 filaments have been reported in highly proliferative cells, such as pluripotent stem cells (Carcamo et al, 2011; Keppeke et al, 2018) and activated T-lymphocytes (Calise et al, 2018; Duong-Ly et al, 2018), and upon pharmacological treatments that block IMP to GTP flux in cell cultures (Ji et al, 2006; Thomas et al, 2012). Our work is the first showing IMPDH2 filaments in postmitotic neurons and in pathological conditions.

Interestingly, we show that only some neurons exposed to AMPD deficiency and GTP depletion assemble IMPDH2 into filaments, suggesting that IMPDH2 polymerization occurs under specific intracellular conditions. Whether these conditions depend on cell-specific energy supply, expression of specific proteins, or posttranslational modifications of IMPDH2 remains to be determined. However, our work suggests several possibilities. First, IMP availability relative to GTP (i.e., IMP/GTP ratio) may be involved in the induction of IMPDH2 polymerization, given that hippocampal IMP/GTP ratio is higher than cortical or cerebellar ratio. In agreement, previous work has shown that increasing concentrations of IMP and ATP induce IMPDH2 polymerization in vitro (Johnson and Kollman, 2020). Second, the preservation of GDP levels in the dKO cortex and cerebellum but not in the dKO hippocampus, may correlate with region-specific accumulation of IMPDH2 filaments or with protection against neurodegeneration through mechanisms that warrant further investigation. An interesting hypothesis arising from this observation is that cortical and cerebellar GDP levels could contribute to neuroprotection by maintaining GTPases in their inactive conformation, thereby directing cellular energy towards pathways involved in neuronal survival. Consistent with this hypothesis, GTPase inhibitors have been shown to protect against neuropathological features of neurodegeneration, including in Alzheimer's disease (Boo et al, 2008; Wang et al, 2009). Finally, neuronal type-specific nucleotide demand and composition may also underly differences in IMPDH2 filament density and neurodegeneration. For instance, neurons in the hippocampal dentate gyrus, where IMPDH2 filaments are scarce, may have lower IMP and ATP availability, making them more vulnerable to GTP deprivation compared to neurons in the CA1-3 hippocampal regions, which are rich in IMPDH2 filaments. Although, current limitations on single-cell resolution metabolomic methods preclude us from testing these hypotheses, the model that emerges from our study suggests that neurons with available guanine nucleotide precursors may survive to AMPD deficiency-associated nucleotide imbalances through IMPDH2 filament assembly to boost GTP synthesis upon cellular or subcellular demand.

In addition, using transmission electron microscopy, we found that IMPDH2 filaments are in close proximity with mitochondria, a primary source of ATP that may favor local conditions for IMPDH2 polymerization regardless of the total cellular ATP levels. Consistently, the association of IMPDH2 filaments with mitochondria has been observed previously (Schiavon et al, 2018). Similarly, a fraction of purinosomes localizes near mitochondria to facilitate access of substrates for the de novo purine biosynthesis (Pareek et al, 2020). Therefore, it is possible that IMPDH2 polymerizes near mitochondria to channel mitochondrial ATP for local guanine nucleotide synthesis, potentially supporting GTP-demanding subcellular processes, such as local protein synthesis, which we previously found to be impaired in PCH9 models (Akizu et al, 2013).

Our work is also consistent with species-specific differences in brain purine nucleotide requirements. Indeed, while hippocampal dentate gyrus degenerates in AMPD deficient mice, there is currently no data showing hippocampal degeneration in human AMPD2 deficiency. Furthermore, the cerebellum, which is hypoplastic at birth in human patients, remains apparently intact in AMPD deficient mice, at least up to their death before postnatal day 21. Much of the cerebellar pathology in human AMPD2 deficiency arises during development (Kortum et al, 2018) and, remarkably, there are marked differences in human and mouse cerebellar development, including the timing and progenitor cell types (Haldipur et al, 2019). These differences may be responsible of disparities between human and mouse cerebellar vulnerability to AMPD deficiency. However, it is also possible that the cerebellum would degenerate if AMPD deficient mice survived past day 21. Indeed, lack of IMPDH2 filaments in the cerebellum predicts its susceptibility to degenerate in AMPD deficient mice. Although another possibility is that IMPDH2 filaments accumulate at older ages in the cerebellum protecting it from degeneration. In support of the latest, IMPDH2 levels are also increased in the dKO cerebellum at p20 and immunostainings show small puncta that may represent a less structurally complex IMPDH2 assemblies. Thus, depleting AMPD selectively in cerebella will be required to study cerebellar requirements for purine nucleotides and IMPDH2 filament assembly in the future. Notably, using a similar strategy, deleting AMPD2/3 in the forebrain specifically, we unexpectedly uncovered that, despite showing low IMPDH2 filament density at p19, the cerebral cortex progressively accumulates IMPDH2 filaments, which likely protects it from degeneration.

Whether IMPDH2 and its polymerization into filaments confers resistance to other neurodegenerative conditions is a question that remains to be determined. Interestingly, emerging data suggest this possibility. Indeed, recent studies implicate de novo IMPDH2 mutations in the etiology of neurodevelopmental disorders (Kuukasjarvi et al, 2021; O'Neill et al, 2023; Zech et al, 2020). Although most of the disease variants impair the sensitivity of IMPDH2 for GTP-mediated inhibition, one of these variants (p.S160del) leads to inability of the protein to assemble into filaments (O'Neill et al, 2023). Therefore, it will be interesting to determine if the patient carrying this variant develops a neurodegenerative condition over time. In contrast, mutations in the retinal paralogue, IMPDH1, which are a major cause of photoreceptor degenerative disorders, suggest the opposite. Some of these mutations impair the disassembly of IMPDH1 filaments upon guanine nucleotide supplementation in vitro. Consequently, these mutant IMPDH1 variants constitutively polymerize when overexpressed in cells (Fernandez-Justel et al, 2019; Keppeke et al, 2023). Furthermore, recent data shows that the long-term overexpression of a constitutively polymerizing IMPDH1 disease variant induces cell death in HEp-2 cells line (Keppeke et al, 2023). Consistently, insoluble nuclear inclusions of IMPDH1 have been recently proposed as a mechanism contributing to age-related dysfunction of dopaminergic neurons of the substantia nigra (Woulfe et al, 2024). These data indicate data constitutive IMPDH2 filament formation may be as detrimental as the inability to form filaments and suggest that the coordination between IMPDH2 activity demand and its polymerization are critical to support cell survival. In agreement, our work shows that despite supporting

PCH9 NPC growth, further induction of IMPDH2 filaments, by IMPDH2-WT overexpression, does not improve the cell growth, likely due to lack of enough IMP to be fueled into IMPDH2 filaments for guanine nucleotide biosynthesis in PCH9 NPCs.

Nonetheless, if IMPDH2 filament assembly prevents neurodegeneration, could we leverage this for treatment? Our current work suggests that in a scenario where GTP precursors (i.e., IMP) are available, IMPDH2 filaments support cell growth. Interestingly, we previously found that the treatment of PCH9 patients' NPCs with 5-Aminoimidazole-4-carboxamide riboside (AICAR) prevents neurodegeneration by increasing de novo IMP synthesis and restoring GTP levels (Akizu et al, 2013). In addition, evidence shows that AICAR induces IMPDH2 polymerization (Schiavon et al, 2018) suggesting it could potentially act as a dual booster rescuing survival of AMPD2-deficient neurons. Furthermore, AICAR enhances exercise endurance and transiently promotes neurogenesis and learning and memory in healthy mice (Guerrieri and van Praag, 2015; Narkar et al, 2008). Yet, there is also evidence showing that chronic AICAR supplementation may elicit neuronal apoptosis, and impair axon growth and synaptic plasticity (Guerrieri and van Praag, 2015; Ma et al, 2014; Williams et al, 2011). Therefore, although AICAR may not be the best therapeutic candidate, our work opens the possibility for IMPDH2 polymerization combined with de novo purine nucleotide stimulation as a therapeutic target for neurodegeneration, at least in PCH9.

# Methods

## Mice

All animal procedures were performed according to NIH guidelines and approved by the Institutional Animal Care and Use Committee (IACUC) at Children's Hospital of Philadelphia. $Ampd2^{-/-}$ and $Ampd3$ first knock outs mice were generated previously (Akizu et al, 2013; Toyama et al, 2012). $Ampd2^{-/-}$ and $Ampd3^{-/-}$ double knock out (dKO) mice were generated by breeding $Ampd2^{-/+};Ampd3^{-/-}$ males with females. Given that $Ampd2^{-/+};Ampd3^{-/-}$ are grossly indistinguishable from wild-type mice (Akizu et al, 2013), these were used as control littermates for all our experiments. For conditional AMPD deficient mice (cdKO), we first excised the trapping cassette of the $Ampd3^{-/-}$ first knock out allele by FRT mediated recombination after breeding it into ACTB:FLPe (Jackson Laboratory # 005703) transgenic mice. Mice carrying Ampd3 conditional allele ($Ampd3^{f/f}$), with loxp sites flanking exon 3, were then breed into $Ampd2^{-/-}$ background and $Emx1:Cre$ knock in mice (Jackson Laboratory #005628) (Gorski et al, 2002). To generate $Ampd2^{-/-}$; $Ampd3^{f/f}$; $Emx1-Cre^{+/-}$ mice (cdKO) for experiments, $Ampd2^{-/-}$; $Ampd3^{f/f}$ mice were bred with $Ampd2^{+/-}$; $Ampd3^{f/f}$; $Emx1-Cre^{+/-}$ mice. Littermates with $Ampd2^{-/-}$; $Ampd3^{f/f}$ or $Ampd2^{+/-}$; $Ampd3^{f/f}$; $Emx1-Cre^{+/-}$ genotypes were used as controls for experiments. Male and female mice were indistinguishably used for experiments. Mice were housed two to five animal per cage with a 12-h light-dark cycle (lights on from 0600 to 1800h) at constant temperature (23 °C) with ad libitum access to food and water. Routine genotyping was performed by tail biopsy and PCR as previously described (Akizu et al, 2013).

## Human neural progenitor cell culturing

Approval to work with previously reported PCH9 iPSCs (Akizu et al, 2013) was obtained by the Children's Hospital of Philadelphia Institutional Review Board. Neural progenitor cells (NPCs) generated from PCH9-1236 patient iPSCs cultured on Matrigel (Corning, #354277) coated dishes with mTSER1 (STEM-Cell Technologies #85850) media as previously described (Akizu et al, 2013). Briefly, embryoid bodies (EBs) were formed by mechanical dissociation of iPSC colonies and plated in neural induction media (DMEM F12, 1x N2, 1x B27, 1 μM LDN and 1 μM SB431542) under rotation at 75 rpm for 9 days. Resultant EBs were then plated on Matrigel-coated dishes in NBF medium (DMEM F12, 1x N2, 1x B27, 20 ng/ml bFGF, 1x Penicillin-Streptomycin). Neural rosettes were visible to pick after 4–6 days, and NPCs dissociated with Accutase (Thermo Fisher, A1110501) and plated on Matrigel with NBF media. Media was replaced every other day and NPCs passaged once a week approximately. All experiments were performed with NPCs at passage 5–8. Cell cultures were tested for mycoplasm by PCR analysis of cell pellets.

## NPC growth curve assessment

NPCs were plated at $15 \times 10^3$ cells/well in 96-well plates and allowed to adhere overnight. For growth curve analysis, viable number of cells were measured with MTT (Sigma-Aldrich M2128-1G) at 0 h (the day after plating), 24 h and 72 h. Briefly, 0.325 mg/ml MTT (3-(4,5-Dimethylthiazol-2-yl)-2,5-Diphenyltetrazo-lium Bromide; Sigma) was added into culture medium for 2 h and incubated at 37 °C. Media was then aspired and DMSO added to lyse NPCs and dissolve the purple formazan crystals formed by the reduction of yellow MTT in viable cells. Absorbance was measured in SpectraMax 190 Microplate reader at 570 nm and subtracted for background signal at 670 nm.

## Transduction of NPCs

NPCs were transduced with lentiviruses carrying Flag, IMPDH2WT-Myc, or IMPDH2p.Y12A-Myc cDNA. To this end, IMPDH2WT-Myc and IMPDH2p.Y12A-Myc in pcDNA3 plasmids (Anthony et al, 2017) (a gift from Jeffrey Peterson) were subcloned into pINDUCER20 (Meerbrey et al, 2011). Lentiviruses were then produced in 10 cm dishes of Hek293T cells, by co-transfection of 10 μg pINDUCER20 plasmids, 5 μg p8.9NdeltaSB (Addgene #132929) and 0.5 μg pCMV-VSV-G (Addgene #8454)) with Lipofectamine 2000 (Invitrogen #11668500). 8 h after transfection, media was replaced with NBF media and 48 h after media with lentiviruses was transferred to NPCs in the presence of 8 mg/ml polybrene. Following 1-week selection with 200 mg/ml G418, NPCs were treated with 100 ng/ml doxycycline for the transgene expression and plated for experiments.

## Immunoblotting

Mice were euthanized with $CO_2$ followed by decapitation, and the region of interest (hippocampus, cortex, and cerebellum) was rapidly dissected and frozen in liquid nitrogen and stored at −80 °C until further processing. Brain and NPC samples were lysed with RIPA buffer (Cell Signaling, 9806S-CST) supplemented with

protease inhibitors (Sigma, P8340). After 30 min of incubation at 4 °C, lysate was centrifuged at 12,000 rpm for 10 min. Protein concentration was determined by Bradford (Thermo Fisher 23246) method according to the manufacturer's instructions. Protein samples were diluted in an equal volume of 2x LDS sample buffer (Thermo Fisher, B0007) and supplemented with DTT to a final concentration of 50 mM (Bio Rad, 1610611). Protein samples (30–50 µg) were separated on 10–18% SDS-PAGE gels and transferred to PVDF membrane, stained with Ponceau S, and blocked with 5% milk in TBS-Tween 20 for 2 h and incubated with primary antibodies diluted in the 5% BSA in TBS-Tween 20 overnight at 4 °C (AMPD2, Sigma HPA045760, 1:1000; IMPDH2, Abcam ab129165, 1:1000; S6, Santa Cruz sc74459, 1:1000; p-S6, Cell signaling 5364, 1:1500; Tubulin, Sigma T6074, 1:2000; Actin, Gen Script A00702 -40, 1:2000). Membranes were washed 3 times with TBS-Tween 20 and incubated with anti-mouse IgG-HRP (Thermo Fisher SA1100; Li-cor Biosciences 926-68072) or anti-rabbit IgG-HRP (Thermo Fisher 31458; Li-cor Biosciences 926-32213) secondary antibodies (all 1:4000) for 2 h at room temperature. After washing 3 times with TBS-Tween 20, membranes were exposed to enhanced chemiluminescence substrate (Thermo Fisher 34076) and signal developed in autoradiography films and AFP Mini-Med 90 X-Ray Film Processor. Bands were quantified using ImageJ (NIH, USA) or Image Studio Lite (USA).

## Histology and immunocytochemistry

Mice were anaesthetized with isoflurane (1–4%) and intracardially perfused with PBS followed by 4% PFA in PBS. Brains were extracted and further fixed in 4% PFA overnight at 4 °C and washed 3 times with PBS. Coronal or sagittal sections of 50 µm were obtained in a vibratome, mounted on slides and dried overnight at room temperature. Dried sections were stained with cresyl violet (Millipore C5042) followed by dehydration in alcohol gradient and clearing with xylene before mounting with Permount (Thermo Fisher SP15) mounting media. Stained brain sections were imaged with an epifluorescent microscope (Leica DM6000). The thickness of hippocampal CA1, CA3, and DG layers was measured using ImageJ Software (NIH, USA) by manually drawing a perpendicular line at three sites along the pyramidal (CA3) or granule (DG) cell layer of each region. The average of the three measurements was used as layer thickness.

For immunofluorescence staining, 50-µm thick brain sections were permeabilized and blocked with blocking solution (5% fetal serum bovine plus 0.5% Triton X-100 in PBS) for 2 h at room temperature. Sections were incubated with primary antibodies (IMPDH2, Abcam ab129165, 1:1000; CD68, Bio Rad MCA1957, 1:250; NEUN, Millipore MAB377, 1:2000; GFAP, Aves Lab 75-240, 1:1000, S6, Santa Cruz sc74459, 1:1000; p-S6, Cell signaling 5364, 1:1500) in blocking solution overnight at 4 °C. Section were washed 3 times with 0.1% Triton X-100 in PBS and incubated with secondary antibodies (Alexa Fluor 488-, 555-, 633-labeled goat anti-mouse, goat anti-rabbit, goat anti-rat or anti-chicken IgGs (H + L) 1:500, Thermo Fisher) and DAPI (1:100, Thermo Fisher, D3571) in blocking solution at room temperature for 2 h. Then, sections were washed 3 times with 0.1% Triton X-100 in PBS and mounted in coverslips Prolong Gold mounting media. Imaging was performed using a Leica SP8 confocal microscope. For z-stack images, 5 µm z-stack confocal images were acquired at 1 µm

intervals. Image processing to adjust for brightness and contrast or change color for consistency across figures was performed using Fiji (ImageJ) Software (NIH, USA). IMPDH2 filament number quantification were made manually, considering only structures larger than 1µm with clear rod and ring shape.

## Transmission electron microscopy (TEM)

Mice were anesthetized with isoflurane (1–4%) and intracardially perfused with 2%PFA + 2% Glutaraldehyde in 0.1 M sodium cacodylate. Before processing for TEM, few hippocampal sections were collected by vibratome and presence of IMPDH2 filaments validated by immunofluorescence staining with anti-IMPDH2. Contiguous hippocampal tissue was post-fixed in 2% osmium tetroxide, dehydrated through a graded ethanol series, infiltrated in propylene oxide and embedded in resin. Semithin sections were stained with toluidine blue and ultrathin sections were stained with lead citrate. Images were captured with a Zeiss Libra I20 TEM. IMPDH2 filaments were identified by comparison with previously published TEM images (Juda et al, 2014; Schiavon et al, 2018) and their absence in control tissue.

## RNA sequencing and data analysis

20-day-old mice were euthanized, and cortex, hippocampus, and cerebellum were dissected on ice, fast frozen, and stored in −80 °C until RNA extraction. On the day of RNA extraction, 50–100 mg tissue from each sample was lysed and extracted in 1 ml TRIzol (Invitrogen, #15596026) according to the manufacturer's recommendations. RNA integrity and quantity were assessed in Bioanalyzer. Strand-specific mRNA-seq libraries were generated at Novogene and sequenced on an Illumina NovaSeq 6000 platform with 2×150 PE configuration at an average of 15 million reads per sample. Adapter and poor quality sequences were trimmed, and cleaned mRNA reads mapped to the *Mus musculus* GRCm38 reference genome using STAR (v2.7.3a) (Dobin et al, 2013). Reads were counted using FeatureCounts from the subread package (v2.0.1) (Liao et al, 2014) with parameters: -p -C -O. Differential gene expression analysis was performed with DEseq2 (V1.38.3) (Love et al, 2014) excluding genes with less than five reads, and all genes from chromosome X and Y. Differential expression was performed separately in each tissue using a linear model and as factor the genotypes. Raw $p$-values were adjusted using the Benjamini-Hochberg method. Differentially expressed genes were defined as having an adjusted $p$ value of less than 0.05. Volcano plots of differentially expressed genes were generated with the EnhancedVolcano R package. Dot plot of gene expression was generated using the tidyverse R package. For Gene Set Enrichment Analysis (GSEA), microglia specific gene sets were generated using the top 500 significantly enriched genes in Microglia_1, Microglia_2, Microglia_3, DAM_Microglia and Perivascular_MF (renamed as Macrophage) (Keren-Shaul et al, 2017). GSEA was performed with the clusterProfiler (Wu et al, 2021) R package using raw $p$-values adjusted with the Benjamini-Hochberg method. GSEA-heatmap was generated using the tidyverse R package. Functional annotation enrichment analysis was performed using Enrichr (Chen et al, 2013) and sub-category MSigDB Hallmark 2000. ENRICHR false discovery rate (FDR) values were calculated using the Benjamini–Hochberg test in ENRICHR and $P$ values were

calculated using Fisher's exact test in ENRICHR. Significantly enriched (pAdj <0.05 ($-$log10 = 1.301) with Benjamini–Hochberg correction) categories were represented as dot plots generated in GraphPad Prism v.10.

## Nucleotide analysis by LC-MS

For mouse brain region nucleotide analysis, mice were intraperitoneally injected with pentobarbital (100 mg/kg) and hippocampus, cortex, and cerebellum were harvested and placed immediately in liquid nitrogen. Frozen brain regions were lyophilized and then homogenized in 80% methanol using a Precellys homogenizer. For NPC cultures, $6 \times 10^6$ NPCs were plated in 10 cm plates and 48 h later, washed 3 times in cold PBS and then fast frozen with liquid nitrogen. Frozen cell lysates and mouse brains were homogenized in ice cold 80% methanol. Aliquots of homogenates were spiked with 10 µL of isotopically-labeled nucleotides as internal standards, extracted with 400 µL of methanol and centrifuged at $18,100 \times g$ for 5 min at 4 °C. Precipitated proteins were quantified using Bradford assay and used for normalization. 400 µL of the supernatant was dried under nitrogen at 45 °C, and reconstituted in HPLC solvents for LC/MS. Calibration solutions (10 µL) and internal standards (10 µL) were spiked into 90 µL of 80% methanol and similarly prepared. Separation and quantitation of nucleotides was achieved with a 9 min linear gradient (95% acetonitrile to 46% acetonitrile with 8.6 min re-equilibration, 0.6 mL/min, 40 °C, 1 µL injection) on a Waters Atlantis Premier HILIC-z column and multiple reaction monitoring of calibration solutions and study samples on an Agilent 1290 Infinity UHPLC/6495 triple quadrupole mass spectrometer. Raw data was processed using Mass Hunter quantitative analysis software (Agilent). Calibration curves (R2 = 0.99 or greater) were either fitted with a linear or a quadratic curve with a 1/X or 1/X2 weighting.

## Statistical analysis

All experiments were performed in at least three independent biological replicates. The $n$ number for each experiment, details of statistical analysis and software are described in the figure legends or main text. Statistical analysis was performed using Prism (GraphPad).

## Data availability

RNAseq data was deposited in GEO under the GSE253045 accession number. All the other data are available in the main text or the supplementary materials.

The source data of this paper are collected in the following database record: biostudies:S-SCDT-10_1038-S44319-024-00218-2.

## Peer review information

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

## Acknowledgements

We thank Shannon Modla at Delaware University Biotechnology Institute electron microscopy core for processing and imaging tissue for transmission electron microscopy. Metabolomics studies were performed by the Penn Metabolomics Core (RRID:SCR_022381) supported by the Penn Cardiovascular Institute and, in part, by NCI P30 CA016520 and NIH P30DK050306. We are also grateful to Jeffrey R. Peterson for IMPDH2-WT and -p.Y12A pCDNA3 plasmids, but especially, for inspiring this work with his insights on IMPDH2 filaments and passion for science. We also thank Teresa Jimenez for technical support during the revision. This work was supported by the National Institute of Health NIH/NINDS R00NS089859 grant (NA) and International Brain Research Organization ISN-Research Postdoctoral Fellowship (MF-M).

## Author contributions

**Marco Flores-Mendez**: Conceptualization; Data curation; Formal analysis; Supervision; Validation; Investigation; Visualization; Methodology; Writing—original draft; Writing—review and editing. **Laura Ohl**: Data curation; Formal analysis; Writing—review and editing. **Thomas Roule**: Software; Formal analysis; Visualization; Writing—review and editing. **Yijing Zhou**: Data curation; Formal analysis; Writing—review and editing. **Jesus A Tintos-Hernández**: Data curation; Formal analysis; Writing—review and editing. **Kelsey Walsh**: Data curation; Writing—review and editing. **Xilma R Ortiz-Gonzalez**: Resources; Supervision; Writing—review and editing. **Naiara Akizu**: Conceptualization; Resources; Data curation; Formal analysis; Supervision; Funding acquisition; Investigation; Visualization; Methodology; Writing—original draft; Project administration; Writing—review and editing.

Source data underlying figure panels in this paper may have individual authorship assigned. Where available, figure panel/source data authorship is listed in the following database record: biostudies:S-SCDT-10_1038-S44319-024-00218-2.

## Disclosure and competing interests statement

The authors declare no competing interests.

# Expanded View Figures

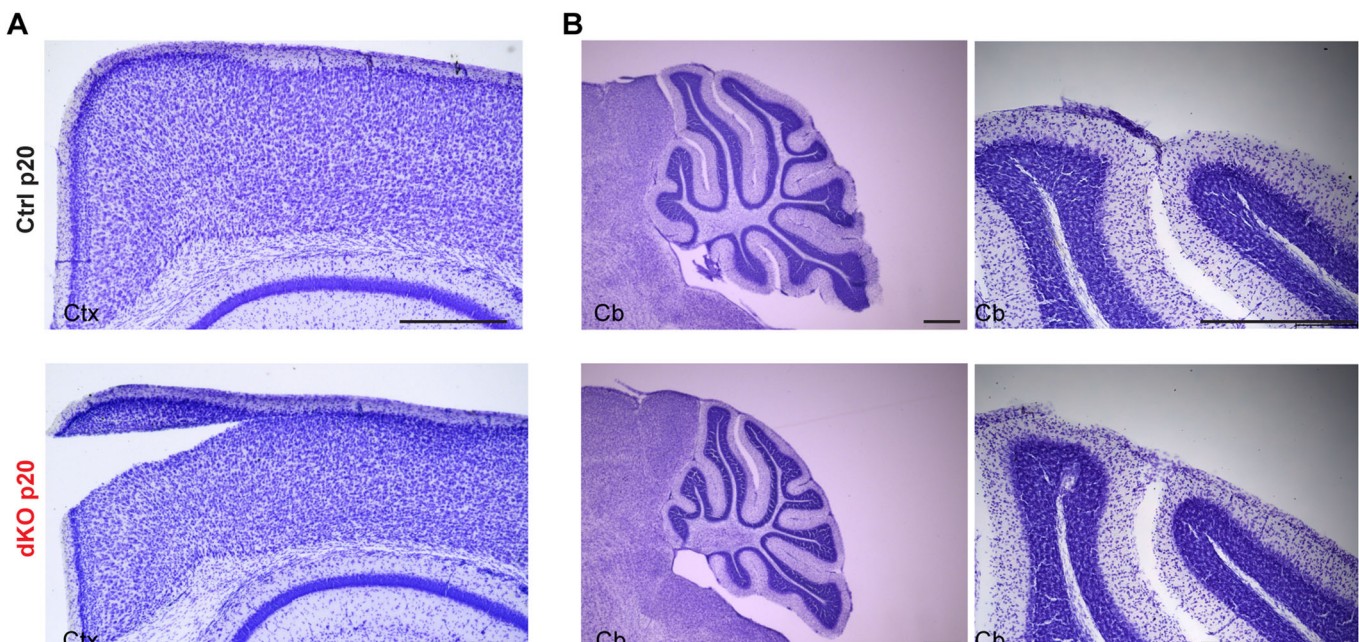

**Figure EV1. No signs of overt neurodegeneration in cortex and cerebellum of Ampd2 and Ampd3 double knockout (dKO) mice.**

(A, B) Representative Nissl staining of the cerebral cortex (A) and cerebellum (B) of control (Ctrl) and double knockout (dKO) mice brain vibratome sections at postnatal day 20 (p20) showing intact tissue, with no structural abnormalities or signs of neurodegeneration. Cerebral cortex images in (A) are magnifications of the whole brain coronal section images shown in Fig. 1E. Ctx = Cortex and Cb = Cerebellum. Scale bars, 500 μm. Source data are available online for this figure.

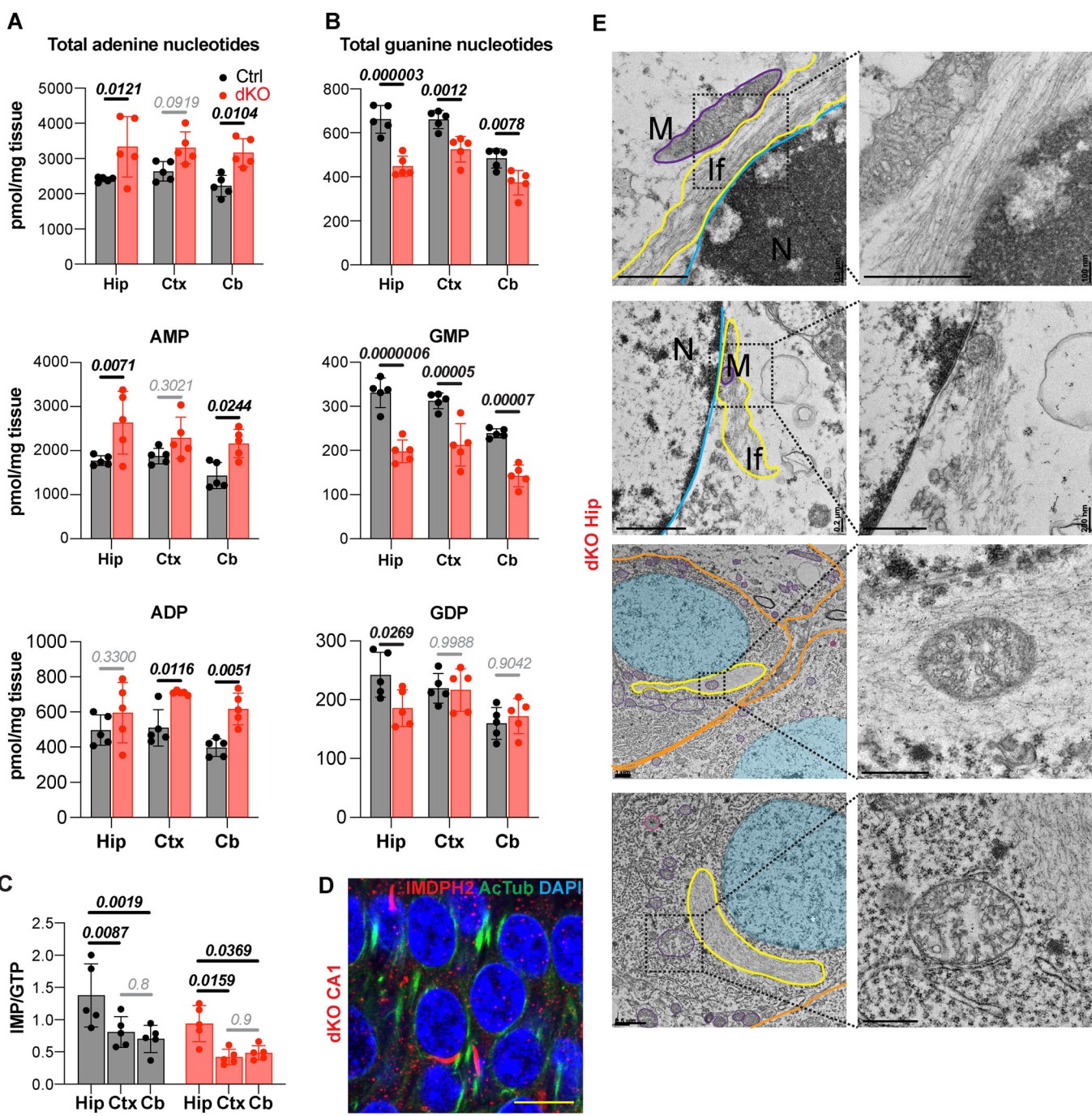

**Figure EV2.  Purine nucleotide imbalance and IMPDH2 filament assembly in dKO mice brain regions.**

(A, B) Graphs showing adenine (A) and guanine (B) nucleotides levels in Hippocampus (Hip), Cortex (Ctx) and Cerebellum (Cb) of control (Ctrl) and double knockout (dKO) mice at postnatal day 20 (p20). Graph shows mean ± SD of $n = 5$ mice per genotype. Significance was calculated using two-way ANOVA with Sidak's post hoc analysis for multiple comparison. (C) Graph showing that the ratio between IMP and GTP levels is larger in control and dKO mice hippocampus (Hip) compared with cortex (Ctx) and cerebellum (Cb). Graphs depict mean ± SD of $n = 5$ mice per genotype. Significance was calculated with two-way ANOVA and Sidak's post hoc analysis for multiple comparison. (D) Representative immunostaining of IMPDH2 and Acetylated tubulin (primary cilia marker) in dKO mouse at p20. Scale bar, 10 μm. (E) Representative transmission electron microscopy images show IMPDH2 filaments (IF) (traced in yellow) within hippocampal neurons. Magnifications (right panels) show IMPDH2 filaments close to mitochondria (M) (traced in purple) and nucleus (N) (blue). Cells are delimited with orange traces. Scale bars, 1 μm (left); Scale bars, 0.5 μm (right). Source data are available online for this figure.

**A**

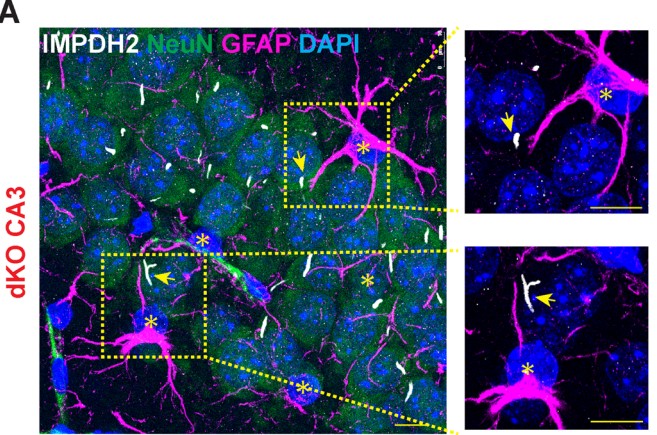

**B**

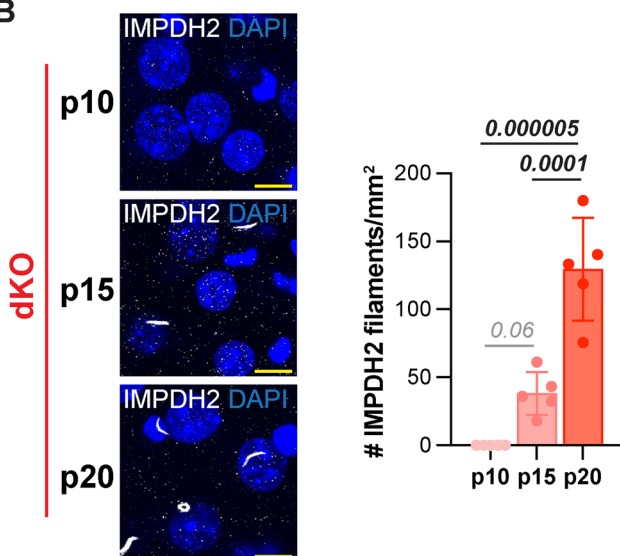

**Figure EV3.    IMPDH2 filaments progressively accumulate in dKO neurons.**

(A) Representative IMPDH2, NEUN (neuronal marker), and GFAP (astrocyte marker) immunostainings showing localization of IMPDH2 filaments predominantly in NEUN+ neurons of dKO mice CA3 region. Yellow arrows point to IMPDH2 filaments in neurons and yellow asterisks to astrocytes. Scale bars, 10 µm. (B) Representative IMPDH2 immunostainings of dKO cortex (Ctx) at postnatal days 10, 15, and 20, showing age-dependent progressive accumulation of IMPDH2 filaments. Scale bars, 10 µm. Graph shows mean ± SD of IMPDH2 filament density at p10, p15, and p20 dKO mice. $n = 5$ mice per genotype. Significance was calculated using one-way ANOVA with Tukey's post hoc analysis for multiple comparison. Source data are available online for this figure.

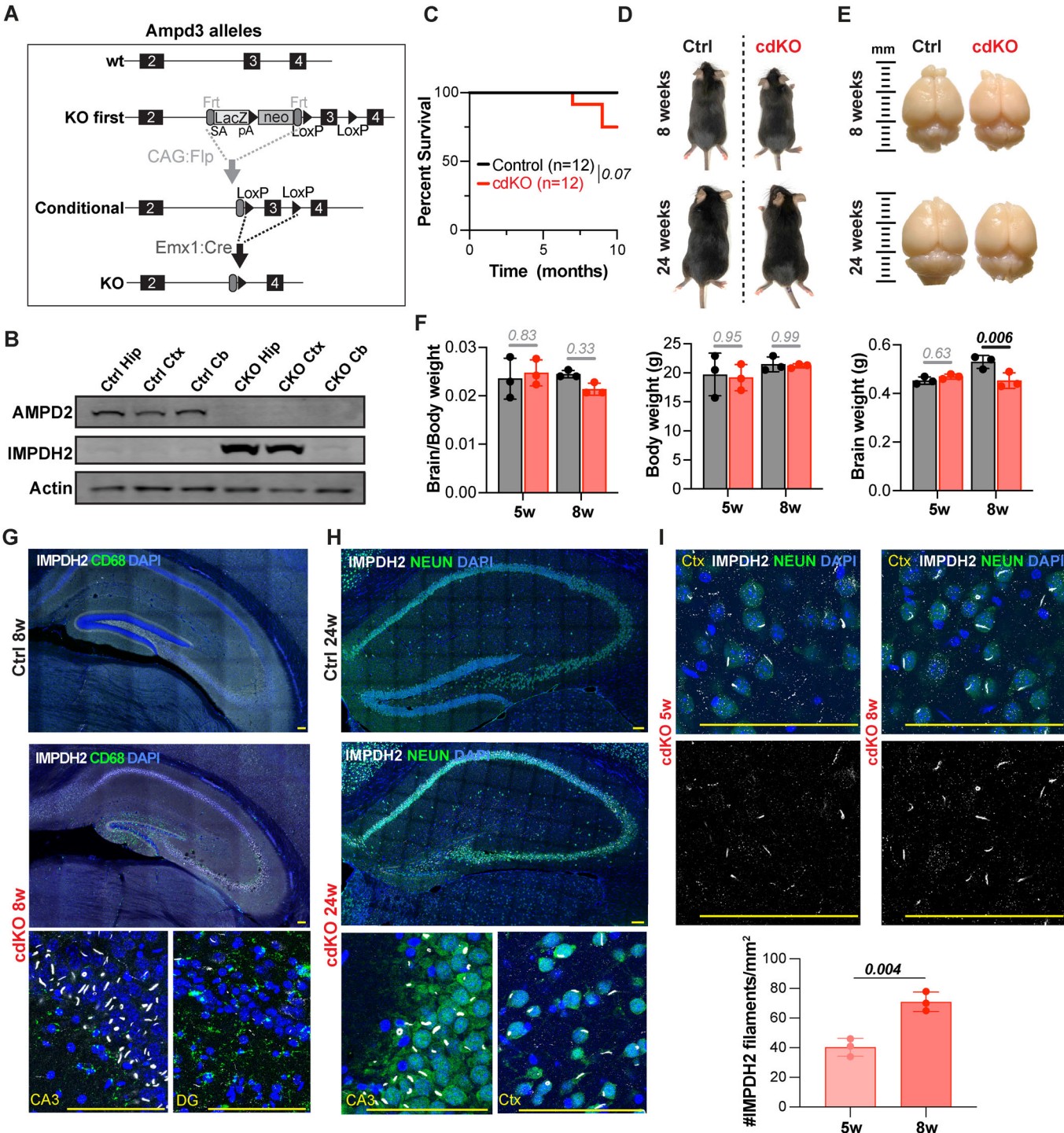

**Figure EV4. Forebrain deletion of *Ampd2* and *Ampd3* leads to mild loss of brain weight, microglia accumulation in hippocampal DG, and progressive IMPDH2 filament accumulation in CA1-3 and cortex.**

(A) Schematic diagram showing Cre-mediated excision of floxed Ampd3 exon 3 to generate the conditional double knockout (cdKO) mouse for *Ampd2* and *Ampd3*. (B) Representative Western blot showing absence of AMPD2 and upregulation of IMPDH2 in cdKO compared with control (Ctrl) mice (5w). (C) Kaplan Meier survival curve showing no differences between $n = 12$ control and $n = 12$ cdKO mice analyzed. Significance was calculated using Log-rank (Mantel-Cox) test. (D) Representative image of Ctrl and cdKO mice at 8 and 24 weeks. (E) Brain images from Ctrl and cdKO mice at 8 and 24 weeks. (F) Brain weight, body weight, and their ratio at 5 and 8 weeks of age. Graph shows mean ± SD of $n = 3$ mice per genotype. Significance was calculated using two-way ANOVA with Sidak's post hoc analysis for multiple comparison. (G) Representative immunostainings of IMPDH2 and CD68 in control and cdKO mice at 8 weeks. Bottom panels show higher magnifications of cdKO CA3 with IMPDH2 filaments and low microglia density, and cdKO DG with no IMPDH2 filaments and high microglia density. (H) Representative immunostainings of IMPDH2 and NEUN (neuronal marker) in control and cdKO mice at 24 weeks showing nearly undetectable DG and intact CA1-3. Bottom panels are magnifications of CA3 and Ctx showing dense IMPDH2 filaments. (I) Representative images of cortex (Ctx) in cdKO mice at 5 (left) and 8 weeks (right) of age showing age-dependent progressive IMPDH2 filament accumulation. Bar graph shows mean ± SD of IMPDH2 filament density in $n = 3$ cdKO mice at 5w and 8w. Significance was calculated using unpaired t-test analysis. Scale bars, 100 μm. Source data are available online for this figure.

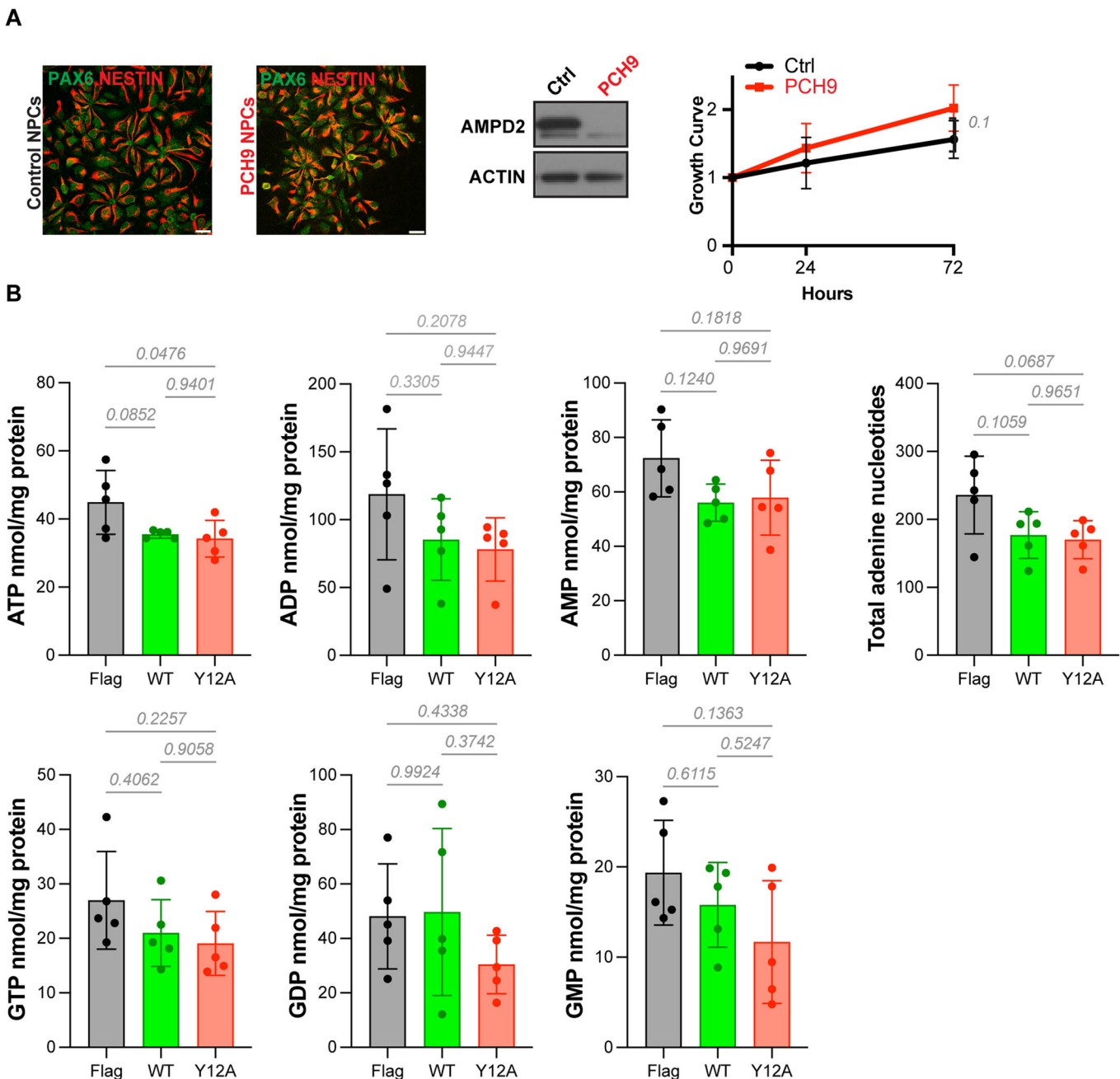

**Figure EV5. Similar purine nucleotide levels upon IMPDH2-WT or IMPDH2-p.Y12A transduction in PCH9 NPCs.**

(A) Representative images of PCH9 patient and unaffected control NPCs immunostained with anti-PAX6 and anti-NESTIN NPC markers. Western Blot analysis showing lack of AMPD2 protein in PCH9 NPCs compared to controls. Graph shows growth curves of control (Ctrl) and PCH9 NPCs over a 72 h period. Graph shows mean ± SD of 4 independent cultures. Statistical difference was calculated comparing slops of simple linear regression. (B) Adenine and Guanine nucleotides levels in PCH9 NPCs expressing FLAG, IMPDH2-WT, and IMPDH2-p.Y12A. Graph shows mean ± SD of $n = 5$ samples per group. Significance was calculated with one-way ANOVA with Tukey's post hoc analysis for multiple comparison. Scale bars, 25 μm. Source data are available online for this figure.

