## [Peer Review File · EMBO Reports]

IMPDH2 filaments protect from neurodegeneration in AMPD2 deficiency

Marco Flores-Mendez, Laura Ohl, Thomas Roule, Yijing Zhou, Jesus Tintos-Hernández, Kelsey Walsh, Xilma Ortiz-Gonzalez, and Naiara Akizu

Corresponding author(s): Naiara Akizu (aquizun@chop.edu)

Review Timeline:

Submission Date:	15th Jan 24
Editorial Decision:	19th Feb 24
Revision Received:	1st Jun 24
Editorial Decision:	1st Jul 24
Revision Received:	6th Jul 24
Accepted:	16th Jul 24

Editor: Esther Schnapp

Transaction Report:

Dear Dr. Akizu,

Thank you for the submission of your manuscript to EMBO reports. We have now received the full set of referee reports that is pasted below.

As you will see, all referees acknowledge that the findings are interesting. They only raise a few concerns and suggest a few more experiments, and I think all suggestions should be addressed. It will be important that the correlative nature of the experiments (where it applies) is clearly stated in the ms text. Please let me know if you have any comments or questions regarding the revisions, and we can discuss this further, also in a video chat, if you like.

I would thus like to invite you to revise your manuscript with the understanding that the referee concerns must be fully addressed and their suggestions taken on board. Please address all referee concerns in a complete point-by-point response. Acceptance of the manuscript will depend on a positive outcome of a second round of review. It is EMBO reports policy to allow a single round of major revision only and acceptance or rejection of the manuscript will therefore depend on the completeness of your responses included in the next, final version of the manuscript.

We realize that it is difficult to revise to a specific deadline. In the interest of protecting the conceptual advance provided by the work, we recommend a revision within 3 months (21st May 2024). Please discuss the revision progress ahead of this time with the editor if you require more time to complete the revisions.

- 1) A data availability section providing access to data deposited in public databases is missing. If you have not deposited any data, please add a sentence to the data availability section that explains that.
- 2) Your manuscript contains statistics and error bars based on $n=2$. Please use scatter blots in these cases. No statistics should be calculated if $n=2$.

5) a complete author checklist, which you can download from our author guidelines <https://www.embopress.org/page/journal/14693178/authorguide>. Please insert information in the checklist that is also

reflected in the manuscript. The completed author checklist will also be part of the RPF.

6) Please note that all corresponding authors are required to supply an ORCID ID for their name upon submission of a revised manuscript (<<https://orcid.org/>>). Please find instructions on how to link your ORCID ID to your account in our manuscript tracking system in our Author guidelines <<https://www.embopress.org/page/journal/14693178/authorguide#authorshipguidelines>>

7) Before submitting your revision, primary datasets produced in this study need to be deposited in an appropriate public database (see <https://www.embopress.org/page/journal/14693178/authorguide#datadeposition>). Please remember to provide a reviewer password if the datasets are not yet public. The accession numbers and database should be listed in a formal "Data Availability" section placed after Materials & Method (see also <https://www.embopress.org/page/journal/14693178/authorguide#datadeposition>). Please note that the Data Availability Section is restricted to new primary data that are part of this study. * Note - All links should resolve to a page where the data can be accessed. *
If your study has not produced novel datasets, please mention this fact in the Data Availability Section.

I look forward to seeing a revised form of your manuscript when it is ready.

Referee #1:

In this study, Flores-Mendez and co-workers demonstrate neurodegeneration confined predominantly to the hippocampus, and specifically to the dentate gyrus of the hippocampus, in mice with a double knockout of the purine biosynthetic enzymes AMPD2 and 3. Mutations in AMPD2 are linked to the human disease PCH9.

They further demonstrate significant GTP depletion in 3 separate areas (hippocampus, cerebellum and cortex) as well as the formation of IMPDH filamentous inclusions in the hippocampus and cortex, but not the cerebellum. The authors attribute the hippocampal neurodegeneration to GTP depletion and correlate the presence of IMPDH filaments with resistance to neurodegeneration, at least in the hippocampus. The data are supported by further studies in mice with a forebrain restricted knockout as well as with neural stem cells from PCH patients and controls. The authors claim that filament formation is neuroprotective in the face of GTP depletion induced by AMPD deficiency. If confirmed, the authors' claims could have significant implications for understanding, and even treating nucleotide-related metabolic dysfunction in pediatric, as well as adult neurodegenerative diseases. The study and the claims are entirely novel and appropriately discussed in the context of the previous literature. This study will be interesting to a broad audience including those interested in neuroscience, neurobiology, and cell biology. The paper is unique in the field of enzyme filamentation in that it is one of the first to study this phenomenon *in vivo* in the brain. My major concerns relate to the interpretation of the findings and the veracity of the claims based on the correlational data, as detailed below:

Major Concerns:

1. How can the absence of filaments in the DG be dissociated from neuronal loss/degeneration? The authors' claims are based on correlational data. The claim that IMPDH filament formation is neuroprotective in AMPD-deficient mice, (presumably due to its ability to offset GTP depletion), is based primarily on the regional distribution of IMPDH filaments in the hippocampus. In P20 mice (one day before death) IMPDH filaments are absent in the dentate gyrus which, in Figure 1D, shows significant degeneration but prominent in other CA sectors that are preserved. The authors conclude that filament formation is protective. However, if the dentate gyrus is degenerated, how do the authors know that it had lots of filaments that disappeared when the neurons degenerated. Going further, although unlikely, the authors have not excluded the possibility that filaments in dentate gyrus cells were plentiful but detrimental; their absence in the P20 dentate gyrus reflects selective loss of filament-containing cells. It would be important to know what the DG looks like prior to degeneration. Perhaps the authors have looked at earlier time points? but if they have it should be included.

2. Related to this, if filament formation is protective, why is there no significant neurodegeneration in the cortex or cerebellum which show GTP depletion, but little or no filament formation? (respectively). The authors seem to attribute this regional vulnerability to neuroinflammation and apoptosis. However, it is possible, and indeed probable, that these are secondary to whatever is causing neuronal loss in the DG.

Having said this, I think this is a valuable paper that should be published. I would simply request that the authors address these caveats in their Discussion.

3. Figure 6E: Shows a significant reduction in growth of the Y12 transduced cells relative to FLAG and no significant difference between FLAG and WT. But the IMPDH2-WT MYC construct is the real control and the authors do not show or state whether there is any difference in growth between IMPDH2WY-MYC and IMPDH2Y12-MYC.

Minor Concerns:

1. The manuscript would benefit from careful proof-reading. There are several examples of dropped articles and a few typos.

2. The rationale for showing the absence of filaments in microglia and astrocytes is uncertain when this is already clearly demonstrated using dual immunofluorescence for IMPDH2 and NeuN. If the paper needs to be shortened, this could be omitted.

3. The authors refer to studies (such as the one by Johnson and Kollman) showing that filament formation resists feedback inhibition by IMPDH, thereby sustaining GTP synthesis under conditions of increased GTP demand. While this applies to

individual filaments, the situation for the mesoscale, microscopically visible filament bundles (cytoophidia) may be more complex. Some statement to this effect is warranted.

4. It may help the reader who is not familiar with nucleotide metabolism to have a simplified schematic focussed on the relevant biochemical pathway? Perhaps in the Supplementary material?

5. Figure 2F. The high magnification inset for the DG is markedly pixelated. Do the authors have a better image?

6. In the Results, at the end of the 2nd paragraph in the section: "IMPDH2 filament accumulation in CA1-3 associates with..." there is reference to Figure S4D which I don't have.

7. Please correct the labelling of images in the legend of Figure 6.

8. Figure S2B shows preservation of GDP in the cortex and cerebellum relative to the hippocampus in dKO mice. Why is this? Could this explain the lack of neurodegeneration in those areas?

Referee #2:

The manuscript from Flores-Mendez et al. tells an intriguing story linking assembly of micron-scale filaments of the nucleotide biosynthesis enzyme IMPDH2 to absence of degeneration in certain regions of the brain in a mouse model of AMPD2 deficiency. The authors use a variety of techniques to systematically investigate the cause of the selective neuronal vulnerability and conclude that IMPDH2 filaments are necessary to maintain the metabolic program required for cells to thrive. They follow up a significant amount of mouse model data with data from cultured neural progenitor cells that demonstrates that the ability to form filaments is important for maintaining cell growth. The models proposed by the authors based on their data fit with existing literature on the function of IMPDH filaments and expand the physiologic relevance of IMPDH filament assembly. Their work should be appealing to a variety of readers interested not only in IMPDH2 but in general neurodegeneration mechanisms. The work may be potentially improved by experiments using gene editing of cultured cells, but those experiments are likely beyond the scope of this current manuscript. I have no reservations about this manuscript being published, but I have listed some areas where the manuscript could be clarified to improve the experience of the reader.

Major comments

1. Please check the references cited in the third paragraph of the Introduction - the references made in regards to IMPDH filaments don't seem to make sense in the context they appear. E.g. Gunter 2008, Ji 2006, and Juda 2014 do not discuss filaments in a physiological context, and it was not discovered that filaments make the enzyme more resistant to GTP inhibition until Buey lab's 2019 paper, so the Buey 2015 reference is incorrect.

2. For Fig. 1A, was any quantification of the body sizes of the dKO mice performed and could that be included in Fig. 1? It may also be useful to mention in the main text exactly how many mice were used (n = ?).

3. Fig. 1D, is it possible to quantify the width of dentate gyrus in dKO vs control mice? It's mentioned they are thinner but by how much?

4. Fig. 2E is a bit hard to see, it may benefit from either zooming in each panel so the IMPDH2 structures are more visible or including some insets with magnified views.

5. Fig. 2E, the main text says there were nearly no IMPDH2 filaments in the cerebellum, but from the figure it appears there may be some dot-like structures? It is hard to tell without zooming in more. Can the authors clarify this - maybe zooming in helps.

6. Fig. 2F, quantification for the cerebellum could be included.

7. Fig. 3H, the main text states "Consistently, the expression levels of several microglia marker genes were also the highest in the dKO hippocampus (Fig 3H)" but the heat map looks pretty similar for the three brain regions. Can the authors clarify which genes specifically they are referencing here?

8. Fig. 4A needs additional labels for the regions of the hippocampus, i.e. DG and CA1-3, etc.

9. Fig. S3C, it is mentioned that the life span of cdKO mice was comparable to control littermates, but the data are not shown - could be included here.

10. Fig 5A, the text says the 5 week old cdKO mice have thinner CA1-3 and DG but from the images in panel A they don't seem to be significantly thinner? Is there any quantification to show?

11. Fig. 6B, IMPDH2 filaments seem sparse in the PCH9 NPCs (or maybe they are just hard to see), what percentage of cells formed filaments in the PCH9 NPCs vs. control cells?
12. Fig. 6E, is it possible to follow cell proliferation beyond 72 hours with these cells? You may see a more convincing effect at 96 hours?
13. Is there a possibility to CRISPR NPCs to insert Y12A mutation into the genome? It may be a great future experiment to show NPCs without wildtype IMPDH1/2 and only Y12A have a severe growth defect. (I am not suggesting this as an experiment for this manuscript, I ask as a curiosity)

Minor comments

1. Fig. 1D, please label the cerebellum and cerebrum to orient non-expert readers.
2. Fig. 2E, it is not clear from the text what NeuN staining is for non-expert readers.
3. Fig. S2E, what is the orange outline?
4. This sentence in the Results section references Fig. S4D but it should say S3D: "Remarkably, only the DG region of the cdKO mice showed a severe tissue loss at 8th week of age (Fig 5B), while the CA1-3 regions remained similar to controls over time (Fig 5B and S4D)."
5. Fig. 6, the figure legend needs to be checked - panel C is referenced twice and panel D reference is to panel E.

Referee #3:

The manuscript by Flores-Mendez et al. "IMPDH2 filaments protect from neurodegeneration in AMPD2 deficiency" reports that in a genetic context of AMPD deficiency in mice that causes a generalized reduction in brain GTP levels, whether a region is more vulnerable (hippocampus dentate gyrus) or more resistant (cortex, hippocampus CA1 and CA3 regions) to GTP deprivation depends on the absence or the accumulation of micron sized filaments of IMPDH2, the rate-limiting enzyme in de novo GTP synthesis. These filaments likely serve to extend the pool of guanine nucleotides as long as there are purine precursors. In brief, the single key finding of this study is that the accumulation of IMPDH2 filaments confers regional neuronal resistance to the GTP reduction caused by ablation of Adenosine Monophosphate Deaminase (AMPD). This finding is significant because *Ampd2*^{-/-}/*Ampd3*^{-/-} (dKO) mice are a model of PCH9 (Pontocerebellar hypoplasia type 9, caused by loss of function mutations in AMPD2), a neurodegenerative pediatric condition. While the dKO mice do not reproduce the clinical signs specifically at the pons or cerebellum, they present a generalized reduction in brain GTP and selective degeneration of the hippocampal dentate gyrus, that are here explored to dissect the mechanism that confers resistance to AMPD loss of function in certain brain regions. The finding that IMPDH2 filaments protect from AMPD deficiency is novel; it is relevant to the PCH9 pediatric condition; and it advances our understanding of how alterations in ubiquitous metabolic pathways affect neural tissue. It should be of interest to a broad audience, and I find it suitable for publication at EMBO Reports.

The finding is solidly substantiated on: i) nucleotide determinations by HPLC-MS in cortex, cerebellum and hippocampal samples of *Ampd2*^{-/-}; *ampd3*^{-/-} mice that show that AMPD deficiency results in a generalized increase in adenine nucleotides and decrease in IMP and guanine nucleotides; ii) the establishment of a robust inverse correlation between the presence and density of IMPDH2 filaments in the cortex, hippocampal CA1, CA3 and DG regions of *Ampd2*^{-/-}; *ampd3*^{-/-} mice and the vulnerability of these regions to GTP deprivation; iii) an RNAseq analysis of these regions that identifies microglia activation as a distinctive feature of the most vulnerable region; iv) the establishment of an inverse correlation between IMPDH2 filament density and microglia infiltration; and positive correlation between microglia infiltration and neurodegeneration in a longitudinal study of a established conditional knock-out model of AMPD in the forebrain; and v) importantly, the cause-effect relationship between IMPDH2 filament formation and neuronal protection is shown on neuronal progenitor cells derived from PCH9 patients by a genetic approach, as authors show that transfection of NPCs with Y12A-IMPDH2, a dominant negative mutant that impairs filament formation, results in NPC growth restrictions.

I have a few major and minor comments that might be taken into consideration for improvement of the manuscript.

Major comments:

1. On a first approximation, the fact that the cerebellum does not present alterations in *Ampd2*/*Ampd3* dKO mice at p20 (therefore being resistant to AMPD loss of function) seems at odds with the main finding of this manuscript (that resistance is conferred by IMPDH2 filament formation) as the cerebellum does not present an accumulation of IMPDH2 filaments. However, this observation is appropriately discussed in Discussion.
2. A determination of nucleotides was performed on hippocampal samples. Individual nucleotide determinations on the CA1, CA3 and dentate gyrus regions of the hippocampus might further substantiate the finding that IMPDH2 filaments confer resistance to GTP deprivation by increasing guanine nucleotides.
3. The same applies to PCH9 NPCs. Nucleotide determinations should confirm that untransfected NPCs, or myc-IMPDH2-

transfected NPCs present higher guanine nt levels than Y12A-IMPDH2-transfected NPCs. [Filament impairment should correlate with a reduction of guanine nt levels].

4. Authors should explain in FigS2E how they unequivocally identify the IMPDH2 filaments in TEM micrographs (how were filaments identified and delimited?).

5. Given that nucleotide determinations by High Pressure Liquid Chromatography and Mass Spectrometry are not technically easy, it would be very valuable if authors contributed a Supplementary Figure with the chromatograms obtained with their method, particularly for tri-phospho nucleotides.

Minor comments:

1. Page 4, subheading: "AMPD deficiency induces IMPDH2 aggregation in the hippocampus", correct typo.

2. Page 10, first paragraph: "These data indicate data constitutive IMPDH1/2 filament", correct typo.

3. Fig. 2 Legend: (F) Representative immunostaining, correct typo.

4. Fig 6 Legend: correct (C), (D), (E).

RESPONSE TO REFEREES

We thank all the referees for their encouraging and insightful feedback, which has contributed to significantly improve our manuscript. Following their suggestions, our revised manuscript includes thickness quantifications of hippocampal regions, longitudinal analyses of IMPDH2 filament density and new nucleotide quantifications among other revised data. Furthermore, we have edited the manuscript to improve figure quality, provide clarity, and correct previous writing and literature citation errors. Below we include our point-by-point response to the referees' comments:

Referee #1:

In this study, Flores-Mendez and co-workers demonstrate neurodegeneration confined predominantly to the hippocampus, and specifically to the dentate gyrus of the hippocampus, in mice with a double knockout of the purine biosynthetic enzymes AMPD2 and 3. Mutations in AMPD2 are linked to the human disease PCH9. They further demonstrate significant GTP depletion in 3 separate areas (hippocampus, cerebellum and cortex) as well as the formation of IMPDH filamentous inclusions in the hippocampus and cortex, but not the cerebellum. The authors attribute the hippocampal neurodegeneration to GTP depletion and correlate the presence of IMPDH filaments with resistance to neurodegeneration, at least in the hippocampus. The data are supported by further studies in mice with a forebrain restricted knockout as well as with neural stem cells from PCH patients and controls. The authors claim that filament formation is neuroprotective in the face of GTP depletion induced by AMPD deficiency. If confirmed, the authors' claims could have significant implications for understanding, and even treating nucleotide-related metabolic dysfunction in pediatric, as well as adult neurodegenerative diseases. The study and the claims are entirely novel and appropriately discussed in the context of the previous literature. This study will be interesting to a broad audience including those interested in neuroscience, neurobiology, and cell biology. The paper is unique in the field of enzyme filamentation in that it is one of the first to study this phenomenon *in vivo* in the brain. My major concerns relate to the interpretation of the findings and the veracity of the claims based on the correlational data, as detailed below:

We thank Referee #1 for noting the relevance of our work and for the suggestions to improve our findings.

Major Concerns:

1. How can the absence of filaments in the DG be dissociated from neuronal loss/degeneration? The authors' claims are based on correlational data. The claim that IMPDH filament formation is neuroprotective in AMPD-deficient mice, (presumably due to its ability to offset GTP depletion), is based primarily on the regional distribution of IMPDH filaments in the hippocampus. In P20 mice (one day before death) IMPDH filaments are absent in the dentate gyrus which, in Figure 1D, shows significant degeneration but prominent in other CA sectors that are preserved. The authors conclude that filament formation is protective. However, if the dentate gyrus is degenerated, how do the authors know that it had lots of filaments that disappeared when the neurons degenerated. Going further, although unlikely, the authors have not excluded the possibility that filaments in dentate gyrus cells were plentiful but detrimental; their absence in the P20 dentate gyrus reflects selective loss of filament-containing cells. It would be important to know what the DG looks like prior to degeneration. Perhaps the authors have looked at earlier time points? but if they have it should be included.

Referee 1# brings up an outstanding question that we originally tried to address with cdKO mice. Given the longer survival of cdKO mice, we anticipated that, if IMPDH2 filaments were deleterious, neurons in the CA regions, which show filaments, would eventually die. However, as shown in Figure 5, despite exhibiting IMPDH2 filaments persistently, CA neurons in cdKO mice remain intact up to 8 weeks of age. Furthermore, in the revised manuscript we include immunofluorescence images of 24 weeks old cdKO and control mouse brain sections (Fig EV4H) showing preservation of CA regions with abundant IMPDH2 filaments. In contrast, DG regions, which show low IMPDH2 density at all analyzed ages (5 and 8 week old in cdKO), progressively degenerate.

Following Referee 1's suggestion, in the revised manuscript we also include IMPDH2 filament analyses in p10 and p15 dKO mice (Fig 4C). Consistent with cdKO data, these results show that DG regions have low density of IMPDH2 filaments at all analyzed ages, while IMPDH2 filaments are already detected at p15 in CA regions. Together, these data suggests that IMPDH2 filaments are associated with neuroprotection, although we agree with Reviewer 1 that these results are still correlative and do not provide causality *in vivo*. To support causality, in Figure 6, we show that IMPDH2 filament disassembly (by overexpression of IMPDH2-p.Y12A) impairs growth of cultured PCH9 human neural progenitors, supporting a neuroprotective role of IMPDH2 filaments, at least in PCH9 human neural progenitors. We also tried to test causality *in vivo* by creating a mouse strain carrying IMPDH2-p.Y12A mutant allele in the past. Unfortunately, after generating this mouse strain we uncovered that IMPDH2-p.Y12A allele is barely expressed (see FigR1 below), which hampered our ability to test the effect of IMPDH2 disassembly, while preserving the catalytic activity, *in vivo*.

2. Related to this, if filament formation is protective, why is there no significant neurodegeneration in the cortex or cerebellum which show GTP depletion, but little or no filament formation? (respectively). The authors seem to attribute this regional vulnerability to neuroinflammation and apoptosis. However, it is possible, and indeed probable, that these are secondary to whatever is causing neuronal loss in the DG.

Having said this, I think this is a valuable paper that should be published. I would simply request that the authors address these caveats in their Discussion.

We thank the referee for raising this point. As we include in the revised discussion, lack of IMPDH2 filaments in the cerebellum may predict cerebellar susceptibility to degenerate. However, given the reduced survival of dKO mice, we would need to selectively delete *Ampd2/3* from the cerebellum to test this hypothesis. Using a similar strategy, in our study we deleted *Ampd2/3* from the forebrain (cerebral cortex and hippocampus), which revealed that the cerebral cortex progressively accumulates IMPDH2 filaments (Fig. EV4H,I). In the revised manuscript, we also show progressive accumulation of IMPDH2 filament densities in the cortex of p10, p15 and p20 dKO mice (Fig. EV3B). This data anticipates that the cortex may progressively acquire protection from neurodegeneration despite showing an IMPDH2 filament density lower than the hippocampal CA1-3 regions in p20 dKO mice. Therefore, we do not discard this as a possibility also for the AMPD deficient mouse cerebellum.

3. Figure 6E: Shows a significant reduction in growth of the Y12 transduced cells relative to FLAG and no significant difference between FLAG and WT. But the IMPDH2-WT MYC construct is the real control and the authors do not show or state whether there is any difference in growth between IMPDH2WY-MYC and IMPDH2Y12-MYC.

The referee is right. We now show that the difference in growth of IMPDH2-WT and IMPDH2-p.Y12A NPCs is statistically significant. In addition, per referee 2's suggestion, we also extend the growth curve analysis in Fig. 6D to 120 hours. However, we consider that the FLAG transduced NPC line is an important control for baseline growth rate of PCH9 NPCs. Without this control, we would not know if the difference between IMPDH2-p.Y12A and IMPDH2-WT would be caused by a detrimental effect of IMPDH2-p.Y12A in PCH9 NPCs, or enhanced growth due to IMPDH2-WT overexpression. Given that there is no difference between FLAG (transduction control) and IMPDH2-WT, we can conclude that IMPDH2-WT expression does not provide an advantage to PCH9 NPCs, likely due to a limited availability of the extra IMPDH2 for IMP (which we previously showed to be reduced in PCH9 NPCs (Akizu *et al*, 2013)), and therefore IMPDH2-p.Y12A expression induced filament disassembly is detrimental for PCH9 NPCs.

Minor Concerns:

1. The manuscript would benefit from careful proof-reading. There are several examples of dropped articles and a few typos.

We have performed exhaustive proof-reading of the manuscript to correct for typos and revise article citations.

2. The rationale for showing the absence of filaments in microglia and astrocytes is uncertain when this is already clearly demonstrated using dual immunofluorescence for IMPDH2 and NeuN. If the paper needs to be shortened, this could be omitted.

We now summarize this finding, but we keep Fig 4 to show that microglia are enriched in IMPDH2 filament free DG region and absent in neurodegeneration protected CA1-3 regions which progressively accumulate IMPDH2 filaments from p10 to p20.

3. The authors refer to studies (such as the one by Johnson and Kollman) showing that filament formation resists feedback inhibition by IMPDH, thereby sustaining GTP synthesis under conditions of increased GTP demand. While this applies to individual filaments, the situation for the mesoscale, microscopically visible filament bundles (cytoophidia) may be more complex. Some statement to this effect is warranted.

We thank Reviewer 1 for this suggestion. We have updated the manuscript with a revised paragraph 3 of the introduction to highlight that *in vitro* evidence show that at individual level, filaments protect IMPDH1/2 from allosteric inhibition by GTP, but the effect of micron-sized filament bundles on IMPDH1/2 enzymatic function is likely more complex.

4. It may help the reader who is not familiar with nucleotide metabolism to have a simplified schematic focussed on the relevant biochemical pathway? Perhaps in the Supplementary material?

We agree, and now include a simplified schematic of purine nucleotide biosynthesis pathway in Fig. 1A.

5. Figure 2F. The high magnification inset for the DG is markedly pixelated. Do the authors have a better image?

Thank you for this feedback. We have included better images of IMPDH2 filaments across all the figures of the revised manuscript. Also note, that we have reorganized the data in figure 2 to show enrichment of IMPDH2 filaments in the hippocampus compared to cortex and cerebellum (Fig 2F) while hippocampal regional differences are now shown in Fig 4C along with progression from p10 to p20.

6. In the Results, at the end of the 2nd paragraph in the section: "IMPDH2 filament accumulation in CA1-3 associates with..." there is reference to Figure S4D which I don't have.

We have corrected the figure citation in the revised manuscript.

7. Please correct the labelling of images in the legend of Figure 6.

This has been corrected.

8. Figure S2B shows preservation of GDP in the cortex and cerebellum relative to the hippocampus in dKO mice. Why is this? Could this explain the lack of neurodegeneration in those areas?

Thank you for this astute observation. To highlight this finding in the revised manuscript, we explicitly describe the result in the 1st paragraph of page 5 and include a comment in the discussion (middle of the third paragraph of the discussion). Indeed, preservation of GDP in the cortex and cerebellum could protect against neurodegeneration through maintaining GTPases in their inactive conformation and fueling the cellular energy towards pathways involved in neuronal survival at expenses of pathways involved in anabolic or other energy consuming reactions. Consistent with this hypothesis, GTPase inhibitors (Fasudil and Y-27631) have been employed successfully in preclinical studies to treat human diseases, including neurological disorders (Lu *et al*, 2009). Furthermore, the inhibition of the Rho GTPase, Rac1 can decrease amyloid precursor protein synthesis (APP) and β -amyloid production ($A\beta$), which are associated with neurodegeneration in Alzheimer's Disease (Boo *et al*, 2008; Wang *et al*, 2009).

We do not know why GDP levels are lower in the hippocampus of dKO mice compared to controls, but preserved in dKO cortex and cerebellum, but this may also be related with regional differences maintaining the balance between guanine nucleotide biosynthesis and the overall activity of GTP hydrolyzing enzymes. These are exciting questions and interesting hypothesis that warrant further investigation in future studies.

Referee #2:

The manuscript from Flores-Mendez *et al*. tells an intriguing story linking assembly of micron-scale filaments of the nucleotide biosynthesis enzyme IMPDH2 to absence of degeneration in certain regions of the brain in a mouse model of AMPD2 deficiency. The authors use a variety of techniques to systematically investigate the cause of the selective neuronal vulnerability and conclude that IMPDH2 filaments are necessary to maintain the metabolic program required for cells to thrive. They follow up a significant amount of mouse model data with data from cultured neural progenitor cells that demonstrates that the ability to form filaments is important for maintaining cell growth. The models proposed by the authors based on their data fit with existing literature on the function of IMPDH filaments and expand the physiologic relevance of IMPDH filament assembly. Their work should be appealing to a variety of readers interested not only in

IMPDH2 but in general neurodegeneration mechanisms. The work may be potentially improved by experiments using gene editing of cultured cells, but those experiments are likely beyond the scope of this current manuscript. I have no reservations about this manuscript being published, but I have listed some areas where the manuscript could be clarified to improve the experience of the reader.

We thank Referee #2 for the detailed review and suggestions of our manuscript. We have revised the manuscript to address all the comments, and included additional information in our point-by-point answers below.

Major comments

1. Please check the references cited in the third paragraph of the Introduction - the references made in regards to IMPDH filaments don't seem to make sense in the context they appear. E.g. Gunter 2008, Ji 2006, and Juda 2014 do not discuss filaments in a physiological context, and it was not discovered that filaments make the enzyme more resistant to GTP inhibition until Buey lab's 2019 paper, so the Buey 2015 reference is incorrect.

We thank Referee 2 for noting the inconsistencies in the description of IMPDH2 filaments and cited articles. Our revised manuscript includes an updated 3rd paragraph of the introduction with corrected citations and an extended and more precise description of the literature about IMPDH1/2 filaments.

2. For Fig. 1A, was any quantification of the body sizes of the dKO mice performed and could that be included in Fig. 1? It may also be useful to mention in the main text exactly how many mice were used (n = ?).

We have now included additional bar graphs with body and brain weights in Fig. 1B-D. The exact number of mice analyzed is also stated in the figure legend (Ctrl: p5 n=7, p10 and p15 n=9, p20 n=10; dKO: p5 n=5, p10 n=6, p15 n=8, and p20 n=9).

3. Fig. 1D, is it possible to quantify the width of dentate gyrus in dKO vs control mice? It's mentioned they are thinner but by how much?

This is an important question. Therefore, we have quantified the thickness of CA1, CA3 and DG layers of the hippocampus and included a bar graph with results in Fig. 1E of the revised manuscript. This quantification shows that only the DG is thinner (by half) in dKO mice compared to controls while CA1 and CA3 remain intact.

4. Fig. 2E is a bit hard to see, it may benefit from either zooming in each panel so the IMPDH2 structures are more visible or including some insets with magnified views.

We thank the referee for this suggestion. We now include zoomed in images of this figure to improve IMPDH2 filaments visualization (Fig. 2F of the revised manuscript).

5. Fig. 2E, the main text says there were nearly no IMPDH2 filaments in the cerebellum, but from the figure it appears there may be some dot-like structures? It is hard to tell without zooming in more. Can the authors clarify this - maybe zooming in helps.

This is an interesting observation that we include in the revised manuscript's discussion section. We agree with the referee that the IMPDH2 immunostainings in the dKO cerebellum show dot-like structures that may represent smaller IMPDH2 assemblies. However, it is hard to tell if these dots are IMPDH2 assemblies or non-specific signal of precipitates given that the IMPDH2 staining of the control cerebellum also shows some dots. It is also possible that these dots, which are more dense and larger in dKO cerebella, represent small IMPDH2 assemblies. This possibility would agree with the increased IMPDH2 expression levels we observe by WB analysis in dKO cerebella as well (Fig. 2D). Note that for IMPDH2 filament quantifications across the manuscript, we only consider IMPDH2 filaments with clear rod or ring shape (which are usually larger ($>1\mu\text{m}$) than the dots in the dKO cerebellum).

6. Fig. 2F, quantification for the cerebellum could be included.

We have now included IMPDH2 filament density quantifications in Fig. 2F bar graph.

7. Fig. 3H, the main text states "Consistently, the expression levels of several microglia marker genes were also the highest in the dKO hippocampus (Fig 3H)" but the heat map looks pretty similar for the three brain regions. Can the authors clarify which genes specifically they are referencing here?

We appreciate this observation. The heatmap in the Fig 3H of the original manuscript depicted the average transcript per million (tpm) values of each gene in logarithmic scale, without statistical information or normalization for fold change comparison. Therefore, we have updated the figure with a dot plot that includes the fold change between dKO and Ctrl gene expression and the statistical significance for selected microglia markers that contribute to the enrichments of inflammatory response and disease associate microglia gene sets.

9. Fig. S3C, it is mentioned that the life span of cdKO mice was comparable to control littermates, but the data are not shown - could be included here.

We now include a Kaplan Meier survival curve in Fig EV4C.

10. Fig 5A, the text says the 5 week old cdKO mice have thinner CA1-3 and DG but from the images in panel A they don't seem to be significantly thinner? Is there any quantification to show?

Thank you for this remark. Fig 5A (and B) of the revised manuscript includes bar graphs with layer thickness quantifications. These quantifications show a small but significant reduction of DG layer thickness in 5 week old cdKO hippocampus and a large reduction at 8 weeks of age.

11. Fig. 6B, IMPDH2 filaments seem sparse in the PCH9 NPCs (or maybe they are just hard to see), what percentage of cells formed filaments in the PCH9 NPCs vs. control cells?

We agree with the referee. The percentage of IMPDH2 filaments in PCH9 NPCs is low ($8.2\% \pm 2.3\text{SD}$), but significantly different to control NPCs, which do not show any IMPDH2 filament. We now include a bar graph with quantifications in Fig 6A. We interpret this low

percentage as the result of a dynamic (and reversible) IMPDH2 filament polymerization in PCH9 NPCs based on guanine nucleotide requirements or phase of the cell cycle.

12. Fig. 6E, is it possible to follow cell proliferation beyond 72 hours with these cells? You may see a more convincing effect at 96 hours?

We thank the referee for this suggestion. The updated growth curve includes a 120 hour time point showing more convincing effects (Fig. 6D).

13. Is there a possibility to CRISPR NPCs to insert Y12A mutation into the genome? It may be a great future experiment to show NPCs without wildtype IMPDH1/2 and only Y12A have a severe growth defect. (I am not suggesting this as an experiment for this manuscript, I ask as a curiosity).

This is a great suggestion! Indeed, in the past, we tried to use CRISPR to generate a mouse strain carrying *Impdh2-p.Y12A* allele to test whether IMPDH2 filaments were protective against neurodegeneration in dKO mice. Unfortunately, after generating the mouse strain (including several backcrosses to remove potential CRISPR off targets), we uncovered that *Impdh2-p.Y12A* allele is transcribed at very low levels (Fig R1). Remarkably, a recent paper reports a similar effect on an *Impdh2-p.Y12C* mutant mouse model (Peng *et al*, 2024). Given that the nucleotide variation (TAC>GCA) is close to the promoter region of *Impdh2* we anticipate that the alternate nucleotide affects promoter strength and transcription of *Impdh2*. Therefore, although homozygous *Impdh2-p.Y12A* are viable, we could not use them to disassemble IMPDH2 filaments in dKO mice without affecting enzymatic levels. This finding precluded us from using the same strategy to generate PCH9 NPCs carrying *Impdh2-p.Y12A* mutant allele and supported the overexpression strategy instead. Perhaps, this challenge will be overcome by introducing an alternate non-polymerizing variant allele (i.e. IMPDH2-p.R356A).

Minor comments

1. Fig. 1D, please label the cerebellum and cerebrum to orient non-expert readers.

All the images in Fig 1D (now Fig. 1E) are from coronal sections of the cerebrum (or forebrain as stated in figure legend). We have labeled the cortex (ctx), the hippocampus (hip) and hippocampal Cornu Ammonis (CA) and Dentate Gyrus (DG) regions in the corresponding panels. In addition, we show representative images of the cerebral cortex (ctx) and cerebellum (cb) with corresponding labels in Fig. EV1.

2. Fig. 2E, it is not clear from the text what NeuN staining is for non-expert readers.

Thank you for this suggestion. We now state that NEUN is a neuronal marker in Fig. 2 legend.

3. Fig. S2E, what is the orange outline?

The orange outline delimits cell periphery. We now include this information in the figure legend.

4. This sentence in the Results section references Fig. S4D but it should say S3D: "Remarkably, only the DG region of the cdKO mice showed a severe tissue loss at 8th week of age (Fig 5B), while the CA1-3 regions remained similar to controls over time (Fig 5B and S4D)."

We have now corrected this error.

5. Fig. 6, the figure legend needs to be checked - panel C is referenced twice and panel D reference is to panel E.

We have updated Fig. 6 and its legend and checked for correct panel references.

Referee #3:

The manuscript by Flores-Mendez et al. "IMPDH2 filaments protect from neurodegeneration in AMPD2 deficiency" reports that in a genetic context of AMPD deficiency in mice that causes a generalized reduction in brain GTP levels, whether a region is more vulnerable (hippocampus dentate gyrus) or more resistant (cortex, hippocampus CA1 and CA3 regions) to GTP deprivation depends on the absence or the accumulation of micron sized filaments of IMPDH2, the rate-limiting enzyme in de novo GTP synthesis. These filaments likely serve to extend the pool of guanine nucleotides as long as there are purine precursors. In brief, the single key finding of this study is that the accumulation of IMPDH2 filaments confers regional neuronal resistance to the GTP reduction caused by ablation of Adenosine Monophosphate Deaminase (AMPD). This finding is significant because *Ampd2*^{-/-} *Ampd3*^{-/-} (dKO) mice are a model of PCH9 (Pontocerebellar hypoplasia type 9, caused by loss of function mutations in AMPD2), a neurodegenerative pediatric condition. While the dKO mice do not reproduce the clinical signs specifically at the pons or cerebellum, they present a generalized reduction in brain GTP and selective degeneration of the hippocampal dentate gyrus, that are here explored to dissect the mechanism that confers resistance to AMPD loss of function in certain brain regions. The finding that IMPDH2 filaments protect from AMPD deficiency is novel; it is relevant to the PCH9 pediatric condition; and it advances our understanding of how alterations in ubiquitous metabolic pathways

affect neural tissue. It should be of interest to a broad audience, and I find it suitable for publication at EMBO Reports.

The finding is solidly substantiated on: i) nucleotide determinations by HPLC-MS in cortex, cerebellum and hippocampal samples of *Ampd2*^{-/-}; *ampd3*^{-/-} mice that show that AMPD deficiency results in a generalized increase in adenine nucleotides and decrease in IMP and guanine nucleotides; ii) the establishment of a robust inverse correlation between the presence and density of IMPDH2 filaments in the cortex, hippocampal CA1, CA3 and DG regions of *Ampd2*^{-/-}; *ampd3*^{-/-} mice and the vulnerability of these regions to GTP deprivation; iii) an RNAseq analysis of these regions that identifies microglia activation as a distinctive feature of the most vulnerable region; iv) the establishment of an inverse correlation between IMPDH2 filament density and microglia infiltration; and positive correlation between microglia infiltration and neurodegeneration in a longitudinal study of a established conditional knock-out model of AMPD in the forebrain; and v) importantly, the cause-effect relationship between IMPDH2 filament formation and neuronal protection is shown on neuronal progenitor cells derived from PCH9 patients by a genetic approach, as authors show that transfection of NPCs with Y12A-IMPDH2, a dominant negative mutant that impairs filament formation, results in NPC growth restrictions. I have a few major and minor comments that might be taken into consideration for improvement of the manuscript.

We appreciate Referee #3's thorough review and insightful suggestions. As detailed below we have updated the manuscript including most of the suggestions, but the nucleotide analysis in hippocampal regions. We hope improvements and accessibility to more sensitive technology in the future will allow us to determine nucleotide levels at cellular and subcellular resolution and thoroughly address the nucleotide composition that associates with IMPDH2 filament formation at cellular level.

Major comments:

1. On a first approximation, the fact that the cerebellum does not present alterations in *Ampd2*/*Ampd3* dKO mice at p20 (therefore being resistant to AMPD loss of function) seems at odds with the main finding of this manuscript (that resistance is conferred by IMPDH2 filament formation) as the cerebellum does not present an accumulation of IMPDH2 filaments. However, this observation is appropriately discussed in Discussion.

We agree and thank you the referee for noting our interpretation of this data in the Discussion. Furthermore, as per referee 1's suggestion, we have extended this part of the discussion further to also include the possibility that IMPDH2 puncta detected in the cerebellum or filament assembly at later ages may protect the dKO cerebellum against neurodegeneration in mice.

2. A determination of nucleotides was performed on hippocampal samples. Individual nucleotide determinations on the CA1, CA3 and dentate gyrus regions of the hippocampus might further substantiate the finding that IMPDH2 filaments confer resistance to GTP deprivation by increasing guanine nucleotides.

We fully agree with this comment. However, limited accessibility to the technology required to perform this experiment hinders our efforts to determine brain region specific nucleotide levels. With our current nucleotide quantification method by LC/MS, we require at least two hippocampi (one mouse) to detect guanine and inosine nucleotides. Given that the best way to dissect CA and DG tissue is with small punchers (smaller than 1mm³ to avoid contamination of surrounding

tissue), and the volume of a mouse hippocampus is at least 10 times larger (15-20 mm³ for an adult mouse), we would need 10 times as many mice as the amount we used for our bulk analyses (50 mice per genotype), which imposes an important limitation on time and resources. As an alternative, we have considered more sensitive quantitative methods, and imaging mass spectrometry methods (i.e. MALDI-mass spectrometry imaging) for detection of nucleotides in intact tissue sections. However, upon consultation with experts in these technologies, some of whom we have collaborated in the past, none of these methods perform well quantifying IMP or tri-phospho nucleotides, especially GTP. Therefore, although we consider this an important question to answer, we are unable to provide this data currently. We hope to gain access to sensitive technology in the near future to study nucleotide levels at cellular and sub-cellular resolution and better understand the nucleotide composition associated with IMPDH2 filament assembly *in vivo*.

3. The same applies to PCH9 NPCs. Nucleotide determinations should confirm that untransfected NPCs, or myc-IMPDH2-transfected NPCs present higher guanine nt levels than Y12A-IMPDH2-transfected NPCs. [Filament impairment should correlate with a reduction of guanine nt levels].

We also agree with this suggestion. To determine whether the growth impairment caused by IMPDH2 filament disassembly upon IMPDH2-p.Y12A overexpression correlates with reduction of guanine nucleotide levels we quantified nucleotides in control (FLAG transduced), IMPDH2-WT and IMPDH2-p.Y12A overexpressing PCH9 NPCs. Results did not detect significant differences between the three conditions, however total guanine nucleotides were lower in IMPDH2-p.Y12A than in control and IMPDH2-p.Y12A conditions in average. Notably, there is a large variability of nucleotide levels between replicates of the same condition, likely due to the inter-replicate variability of transgene expression efficiency, which may compromise the statistical power of our experiment. Alternatively, it is possible that global guanine nucleotides levels are not involved in the protection provided by IMPDH2 filaments, or that IMPDH2 disassembly affects only nucleotides in subcellular compartments or a small percentage of NPCs, rendering bulk guanine nucleotide extract of PCH9 NPC nearly intact. As for the previous suggestion, we hope that single-cell resolution metabolomic technologies will allow us to address these questions further in the future.

4. Authors should explain in FigS2E how they unequivocally identify the IMPDH2 filaments in TEM micrographs (how were filaments identified and delimited?).

Before processing for transmission electron microscopy (TEM), few hippocampal sections were collected by vibratome sectioning and the presence of IMPDH2 filaments validated by immunofluorescence staining with anti-IMPDH2. Contiguous hippocampal sections were then processed for TEM. IMPDH2 filaments were identified in TEM images by comparison with previously published TEM images (Juda *et al*, 2014; Schiavon *et al*, 2018) (Fig R2) and their absence in control tissue. We now include this information in the Materials and Methods section for TEM.

5. Given that nucleotide determinations by High Pressure Liquid Chromatography and Mass Spectrometry are not technically easy, it would be very valuable if authors contributed a Supplementary Figure with the chromatograms obtained with their method, particularly for tri-phospho nucleotides.

We appreciate this insight and suggestion. We now include Appendix Figure S1 and S2 with representative images of mass chromatograms extracted by Mass Hunter quantitative analysis software for each nucleotide in an LC/MS run. We have also included a more detailed description of the nucleotide quantification method in the manuscript's methods section. Note that this is a standard method for nucleotide analysis used routinely in the metabolomics core at the University of Pennsylvania.

Minor comments:

1. Page 4, subheading: "AMPD deficiency induces IMPHD2 aggregation in the hippocampus", correct typo.

This has been corrected.

2. Page 10, first paragraph: "These data indicate data constitutive IMPDH1/2 filament", correct typo.

We have corrected this typo.

3. Fig. 2 Legend: (F) Representative immunostaining, correct typo.

This has been corrected.

4. Fig 6 Legend: correct (C), (D), (E).

The figure has been modified and panel citation in the legend revised. Thank you for noting these errors.

References

- Akizu N, Cantagrel V, Schroth J, Cai N, Vaux K, McCloskey D, Naviaux RK, Van Vleet J, Fenstermaker AG, Silhavy JL *et al* (2013) AMPD2 regulates GTP synthesis and is mutated in a potentially treatable neurodegenerative brainstem disorder. *Cell* 154: 505-517
- Boo JH, Sohn JH, Kim JE, Song H, Mook-Jung I (2008) Rac1 changes the substrate specificity of gamma-secretase between amyloid precursor protein and Notch1. *Biochem Biophys Res Commun* 372: 913-917
- Juda P, Smigova J, Kovacik L, Bartova E, Raska I (2014) Ultrastructure of cytoplasmic and nuclear inosine-5'-monophosphate dehydrogenase 2 "rods and rings" inclusions. *J Histochem Cytochem* 62: 739-750
- Lu Q, Longo FM, Zhou H, Massa SM, Chen YH (2009) Signaling through Rho GTPase pathway as viable drug target. *Curr Med Chem* 16: 1355-1365
- Peng M, Keppeke GD, Tsai LK, Chang CC, Liu JL, Sung LY (2024) The IMPDH cytoophidium couples metabolism and fetal development in mice. *Cell Mol Life Sci* 81: 210
- Schiavon CR, Griffin ME, Pirozzi M, Parashuraman R, Zhou W, Jinnah HA, Reines D, Kahn RA (2018) Compositional complexity of rods and rings. *Mol Biol Cell* 29: 2303-2316
- Wang PL, Niidome T, Akaike A, Kihara T, Sugimoto H (2009) Rac1 inhibition negatively regulates transcriptional activity of the amyloid precursor protein gene. *J Neurosci Res* 87: 2105-2114

Dear Dr. Akizu,

Thank you for the submission of your revised manuscript. We have now received the enclosed reports from the referees and I am happy to say that all support its publication now.

Only a few editorial requests will need to be addressed before we can proceed with the official acceptance of your manuscript:

- Please upload all main and all EV figures as individual figure files and do not include the figures in the ms file.
- Please add up to 5 keywords to the ms file.
- Please move the Data Availability Section to before the Acknowledgements.
- Please correct the conflict of interest subheading to "Disclosure and Competing Interests Statement"
- The author credits need to be removed from the ms file. All credits need to be entered during online ms submission.
- The following funding info is not entered in our online submission system: - Metabolomics studies were performed by the Penn Metabolomics Core (RRID:SCR_022381) supported by the Penn Cardiovascular Institute and, in part, by NCI P30 CA016520 and NIH P30DK050306. Please add.
- Please add missing callouts for Appendix Figure S1 and Appendix Figure S2 to the ms text.
- Dataset EV1 has the wrong legend (Dataset EV3), please correct. Can you please also explain again whether your Datasets are source data for specific figures? IF this is the case, they should not be called Datasets but source data.
- The APPENDIX file table of content is missing page numbers, please add.
- The title "Supplementary Figures" in the ms file needs to be corrected to "Expanded View Figures"
- Fig 1E and EV1A seem to show the same images in 2 of the panels. Is this correct? If yes, this image re-use needs to be explained in the figure legend.
- Please address the following comments by our data editors:
 1. Please note that the exact p values are not provided in the legends of figures 1d; 2b, f; 3e; 4c; 5e; 6a, d; EV 2b; EV 3b.
 2. Please indicate the statistical test used for data analysis in the legends of figures 3a-d, f, h.
 3. Please note that information related to n is missing in the legend of figure 6a.
 4. Please note that the error bars are not defined in the legend of figure 6a.
 5. Please note that scale bar and its definition are missing for figure EV 2d.

EMBO press papers are accompanied online by A) a short (1-2 sentences) summary of the findings and their significance, B) 2-3 bullet points highlighting key results and C) a synopsis image that is exactly 550 pixels wide and 200-600 pixels high (the height is variable). The synopsis image should provide a sketch of the major findings, like a graphical abstract. Please note that text needs to be readable at the final size. Please send us this information along with the final manuscript.

Referee #1:

The authors have addressed my major and minor concerns satisfactorily.

Referee #2:

The revised manuscript submitted by the authors has significantly improved. The authors carefully addressed all reviewer comments/suggestions and I have no further inquiries. I see no reason why this manuscript should not be published. Congratulations to the authors on their excellent work.

Referee #3:

The authors have made a commendable effort to carefully address all of the reviewers' comments, and the article is in its current form a great contribution to EMBO reports.

All editorial and formatting issues were resolved by the authors.

Naiara Akizu
Children's Hospital of Philadelphia
3501 Civic Center Boulevard
Philadelphia, PA 19104
United States

Dear Dr. Akizu,

I am very pleased to accept your manuscript for publication in the next available issue of EMBO reports. Thank you for your contribution to our journal.
